# In-plume and out-of-plume analysis of aerosol-cloud interactions derived from the 2014-15 Holuhraun volcanic eruption

Amy H. Peace[1,3], Ying Chen[2], George Jordan[3], Daniel G. Partridge[1], Florent Malavelle[3], Eliza Duncan[1], and Jim M. Haywood[1,3]

[1]Faculty of Environment, Science and Economy, University of Exeter, Exeter, EX4 4QE, UK
[2]School of Geography, Earth and Environmental Sciences, University of Birmingham, Birmingham, B15 2TT, UK
[3]Met Office, Exeter, EX1 3PB, UK

*Correspondence to*: Amy H. Peace (amy.peace@metoffice.gov.uk)

**Abstract.** Aerosol effective radiative forcing (ERF) has persisted as the most uncertain aspect of anthropogenic forcing over
the industrial period, limiting our ability to constrain estimates of climate sensitivity and to confidently predict 21st century climate change. Aerosol-cloud interactions are the most uncertain component of aerosol ERF. The 2014-15 Holuhraun volcanic eruption acted as large source of sulphur dioxide, providing an opportunistic experiment for studying aerosol-cloud interactions at a climatically relevant scale. We evaluate the observed aerosol-induced perturbation to marine liquid cloud properties inside the volcanic plume in the first month of the eruption and compare the results to those from UKESM1 (UK Earth System
Model). In the first two weeks, as expected, we find an in-plume shift to smaller and more numerous cloud droplets in both the observations and the simulations. We find an observed increase liquid water path (LWP) values inside the plume that is not captured in UKESM1. However, in the third week, the in-plume shift to smaller and more numerous cloud droplets is neither observed nor modelled, and there are discrepancies between the observed and modelled response in the fourth week. Analysis of the model simulations and trajectory modelling reveals that airmass history and background meteorological factors
can strongly influence aerosol-cloud interactions between the weeks of our analysis. Overall, our study supports the findings of many previous studies; that the aerosol impact on cloud effective radius is significant, with differences in the observed and modelled response on in-cloud LWP.

# 1 Introduction

The evolution of aerosol emissions is thought to have profoundly impacted climate over the industrial period. Increasing emissions of anthropogenic aerosols and their gaseous precursors has exerted a negative radiative forcing on the climate system through the interaction of aerosols with clouds and radiation (Bellouin et al., 2020). The negative radiative forcing of aerosols has masked a proportion of warming from rising greenhouse gas emissions (Eyring et al., 2021), and led to large-scale changes in the water cycle and atmospheric circulation (Douville et al., 2021). Over the coming decades reductions in anthropogenic aerosol emissions are expected due to more ambitious climate change and air quality mitigation policies (Rao et al., 2017). Despite the importance of aerosol-climate interactions, aerosol radiative forcing is the most uncertain component of anthropogenic radiative forcing over the industrial period (Forster et al., 2021). The uncertainty in the magnitude of aerosol radiative forcing impacts the accuracy in which we can project near-term future climate changes (Andreae et al., 2005; Seinfeld et al., 2016; Peace et al., 2020; Watson-Parris and Smith, 2022). Aerosol-cloud interactions (ACI) make up the largest component of the uncertainty in aerosol radiative forcing (Bellouin et al., 2020). It is therefore an important task to continue to improve our understanding of ACI to predict future climate change more confidently.

Marine low-level liquid clouds strongly reflect shortwave radiation. Only small changes in their properties can have a significant impact of the radiative balance of the Earth system (Wood, 2012). Understanding how aerosols modify the properties of these clouds has therefore been the focus of much research. Conceptually, aerosols modify the properties of clouds through a chain of events (e.g. Haywood and Boucher, 2000). Firstly, aerosols act as cloud condensation nuclei (CCN). An increase in aerosol leads to an increase in cloud droplet number concentrations ($N_d$), and for a constant amount of cloud water, a reduction in cloud droplet effective radius ($r_{eff}$). Smaller and more numerous cloud droplets increase the albedo of clouds (Twomey, 1974). These effects have been widely observed (e.g. Bréon et al., 2002; Feingold et al., 2003). An increase in $N_d$ may initiate further adjustments to cloud properties, such as changes in liquid water path (LWP) and cloud fraction, although bidirectional responses in LWP to an increase in $N_d$ have been observed (e.g. Toll et al., 2019) and simulated (e.g. Ackerman et al., 2004). The directionality of the LWP response likely depends on the meteorological conditions present and accordingly whether smaller cloud droplets lead to precipitation suppression which can potentially increase LWP (Albrecht, 1989; Pincus and Baker, 1994), or if the smaller droplets lead to enhanced evaporation and decreased sedimentation which can enhance entrainment and decrease LWP (Ackerman et al., 2004; Bretherton et al., 2007). Recent research has shown significant cancellation of the positive and negative LWP responses is likely at large scales resulting in a weak LWP response to increased aerosol globally (Toll et al., 2019). However, global climate models (GCMs) can disagree with evidence from observations and higher resolution models on the magnitude and sign of the LWP response to increased $N_d$ (Toll et al., 2017; Gryspeerdt et al., 2019). The uncertain response of LWP to increased $N_d$ demonstrates why cloud adjustments to an increase in $N_d$ remain poorly constrained, despite being able to enhance or counteract an increase in cloud albedo due to an increase in smaller cloud droplets.

'Opportunistic' experiments offer a way to improve our understanding of aerosol-cloud interactions in a system where both the aerosol-perturbed and unperturbed background cloud state are reasonably well established (Christensen et al., 2022). The magnitude and sign of ACI can depend on numerous factors including background aerosol concentrations, meteorology and cloud properties (e.g. Stevens and Feingold, 2009; Carslaw et al., 2013). Opportunistic experiments can therefore provide a way to isolate ACI in environments with similar conditions or provide insight into how background conditions affect ACI. Key opportunistic experiments that have been used to study ACI include ship tracks, industrial plumes, wildfires and volcanic eruptions (e.g. Malavelle et al., 2017; Toll et al., 2017; Christensen et al., 2022). In this study, we utilise the 2014-15 Holuhraun effusive volcanic eruption as an opportunistic experiment to assess and improve our understanding of ACI.

The 2014-15 Holuhraun eruption in Iceland (64.85°N, 16.83°W) began on 31st August 2014 and ended on 27th February 2015. This eruption was one of the largest sources of tropospheric volcanic emissions since the 1783-1784 Laki eruption (Ilyinskaya et al., 2017). Ground-based and satellite observations show that the Holuhraun eruption emitted large amounts of $SO_2$ (up to ~100 kt $SO_2$ day$^{-1}$) into the troposphere (Pfeffer et al., 2018; Carboni et al., 2019). The daily $SO_2$ emitted from the eruption was at least a factor of 3 larger than anthropogenic emissions from the whole of Europe (Schmidt et al., 2015). Once emitted, $SO_2$ is readily oxidised into sulphate aerosol, therefore, the Holuhraun eruption created a large aerosol plume. As a result, the 2014-15 Holuhraun eruption provides an opportunistic experiment to investigate ACI hypotheses at a large, climatically relevant scale.

A handful of studies have leveraged the Holuhraun eruption to study ACI using differing approaches. Malavelle et al., (2017) used a climatological approach to identify aerosol-cloud interactions following the eruption. Their results showed a decrease in $r_{eff}$ during October 2014 in both satellite observations and climate model simulations compared to the climatological mean. Yet, satellite observations revealed no clear perturbation to LWP or cloud fraction, unlike climate model responses showing varying LWP changes. Chen et al., (2022) used a machine learning approach to predict the cloud properties that would be expected for September and October 2014 without the presence of the volcanic eruption, given the meteorological conditions. The predicted cloud properties were then compared to satellite observations to isolate the aerosol perturbation to cloud properties following the eruption. Similarly to the climatological approach of Malavelle et al. (2017), the machine learning approach isolated a decrease in $r_{eff}$ but no detectable change in LWP. However, the machine learning approach revealed an aerosol-induced increase in cloud fraction. Lastly, Haghighatnasab et al., (2022) focused on the first week following the eruption, comparing cloud properties inside and outside the $SO_2$ eruption plume in satellite observations and a high-resolution model. This plume analysis approach showed an increase in $N_d$ and decrease in $r_{eff}$ inside the eruption plume in line with the results from Malavelle et al. (2017) and Chen et al., (2022). However, Haghighatnasab et al., (2022) show an observed shift in the distribution of in-plume LWP values, with a decreased likelihood of low LWP values and an increased likelihood of higher LWP values, which is further exaggerated in the high-resolution model.

Our study builds on these previous analyses of aerosol-cloud interactions derived for September 2014. We use satellite observations of aerosol and cloud properties to evaluate the observed ACI following the start of the volcanic eruption and compare our results to simulations from UKESM1 (UK Earth System Model). We add to the plume analysis approach utilised in Haghighatnasab et al., (2022) by using a more detailed plume masking method that isolates areas close to the plume that are likely to be more representative of the cloud fields being perturbed. We also extend the plume analysis from the first week of September 2014 that was analysed in Haghighatnasab et al., (2022) to the rest of the month. The eruption was at its most powerful in September 2014 with large amounts of $SO_2$ released that then reduced during October 2014 (Carboni et al., 2019). The 4-week time period allows us to investigate how airmass history and background meteorological factors influence aerosol-cloud interactions between the weeks of our analysis using the HYSPLIT trajectory model (The Hybrid Single-Particle Lagrangian Integrated Trajectory model). A week-by-week analysis is performed showing that the aerosol conditions in the first two weeks and the last week of September are close to pristine, but during the third week, the background aerosol is significantly perturbed owing to airmass trajectories originating over continental Europe. This breakdown into weeks provides a convenient framework for developing statistical analyses over the month.

## 2 Data and methods

### 2.1 Defining a plume mask from satellite observation of $SO_2$

We use the column amount of $SO_2$ in the lower troposphere to define a plume mask that is used to compare cloud properties inside and outside of the aerosol plume following the eruption.

We obtain the $SO_2$ data product from the Ozone Mapping and Profiler Suite (OMPS) Nadir Mapper (NM) onboard the NASA-NOAA Suomi National Polar-orbiting partnership (SNPP) satellite that was launched in October 2011 (Flynn et al., 2014; Seftor et al., 2014). The Nadir Mapper is a UV spectrometer that measures backscattered solar UV radiance from the Earth and solar irradiance. $SO_2$ absorbs strongly in the UV and therefore the vertical column density of $SO_2$ can be retrieved from satellite measurements of the UV spectrum. The column amount of $SO_2$ is retrieved from OMPS using a principal component analysis (PCA) algorithm (Li et al., 2017, 2020b). We use V2.0 of the $SO_2$ data product in our analysis (NMSO2_PCA_L2 V2.0) (Li et al., 2020a).

The PCA algorithm provides six estimates of the total $SO_2$ vertical column density based on a priori profiles of the centre of mass altitude (Li et al., 2020a). We use the data product that is based on an $SO_2$ plume height in the lower troposphere (TRL) at 3 km, which is a typical height of volcanic degassing and moderate eruptions. Carboni et al., (2019) showed the altitude of the centre of mass of the $SO_2$ Holuhraun eruption plume was mainly confined to within 0-6 km. Following the OMPS quality control procedure, pixels near the edge of the swath and where the solar zenith angle (SZA) > 70° are excluded. OMPS has a nadir resolution of 50 x 50 km and crosses the equator about 13:30 local time. We resample swath data to a regular grid with

resolution of 1.0 x 1.0° using a nearest neighbour method. 1.0 x 1.0° is the same resolution as the dataset of cloud property observations that we use. When creating the plume mask for use with the model simulations, we first re-grid the 1.0 x 1.0° OMPS data to the coarser resolution of the model simulations. The OMPS $SO_2$ vertical column density is unavailable 1 day in each week and we exclude these dates from our analysis. We apply the following analysis in a "Holuhraun" domain of longitude 45°W to 30°E and latitude of 45°N to 80°N (e.g. as in Figure 1).

After processing the $SO_2$ data product to gridded data, the next step in our analysis is to define a suitable plume mask and bounding region around the plume to use in isolating in-plume versus out-of-plume cloud properties. We use a threshold exceedance approach to define the eruption plume mask. We define grid cells where the total column amount of $SO_2 > 1$ DU as being in-plume. This masking approach and threshold exceedance choice was also used in Haghighatnasab et al., (2022). Next, each day we define a bounding box around the plume as the minimum to maximum latitude and longitude of the plume extent. We use this bounding box approach rather than using the whole domain to minimise differences in meteorological conditions between inside and outside the plume, which can confound the aerosol effect on cloud properties (e.g. McCoy et al., 2020). The plume mask and bounding region for each day is shown in animation S1.

## 2.2 Satellite observations of $SO_2$ plume height

Nadir spectrometer instruments in the ultraviolet and infrared can be used to deduce information on $SO_2$ plume altitude (Carboni et al., 2016). The Infrared Atmospheric Sounding Interferometer (IASI) is a Fourier transform interferometer on board the MetOp-A and -B satellites. $SO_2$ height information can be obtained from IASI through the optimal estimation retrieval scheme as explained in Carboni et al., (2012, 2016). In the algorithm, retrievals are performed when detection of $SO_2$ is above a given threshold. The threshold defined for the Holuhraun eruption is 0.49 effective DU (Carboni et al. 2019). The retrieval algorithm determines $SO_2$ column amount and the altitude (mean of a Gaussian profile) of the $SO_2$ plume. We use the output from the IASI retrieval to compare the height of the volcanic $SO_2$ plume against cloud top height.

## 2.3 Satellite observations of cloud properties

We use products of the MODerate resolution Imaging Spectroradiometer (MODIS) onboard the polar-orbiting Aqua and Terra satellites (Platnick et al., 2015) to evaluate perturbations to cloud properties inside the $SO_2$ plume as determined in Section 2.1. We use the MODIS COSP Level 3 daily (MCD06COSP) dataset that combines pixel-scale observations from Terra (MODO6_L2) and Aqua (MYDO6L2) to a regular 1 x 1 ° grid (Pincus et al., 2023). We use the mean of the sampled Level 2 pixels in each Level 3 grid. The dataset was recently produced to facilitate comparison with results from the COSP (CFMIP Observation Simulator Package) MODIS simulator that is a software tool that can be employed in climate models to produce data comparable to satellite observations. The definitions of variables within this dataset are more in line with the MODIS simulator than standard MODIS products. Therefore, the MODIS COSP dataset is particularly useful for observation-model comparison. We analyse marine liquid $N_d$, $r_{eff}$, in-cloud LWP and cloud fraction. In the Level 2 MODIS products, $r_{eff}$, cloud

water path and cloud optical thickness are retrieved from observed multispectral reflectances using a radiative transfer model at 1 km nadir resolution. Cloud phase is retrieved through the phase retrieval algorithm at 1 km resolution (Platnick et al., 2017).

We derive liquid $N_d$ from liquid cloud $r_{eff}$ and cloud optical thickness ($\tau_c$) assuming an adiabatic cloud:

$$N_d = \alpha \tau_c^{0.5} r_{eff}^{-2.5} \tag{1}$$

Where, $\alpha$ is 1.375 $10^{-5}$ m$^{-0.5}$. Only data pixels where cloud optical thickness is between 4 and 70, and $r_{eff}$ between 4 and 30 µm are retained where the retrieval is the most reliable (Quaas et al., 2006), but $N_d$ derived in this way is still subject to uncertainties related to the cloud adiabaticity assumption and uncertainty in underlying cloud property retrievals (Gryspeerdt et al., 2022). We use the liquid cloud retrieval fraction rather than cloud mask fraction to study cloud fraction. The cloud retrieval fraction is lower than the cloud mask fraction in most regions as it excludes pixels identified as sunglint, heavy aerosol or partly cloudy

(Pincus et al, 2023).

In addition, we use the Level 2 Collection 6.1 MODIS Aqua products (Platnick et al, 2015; Platnick et al., 2017) sampled to a 0.5 x 0.5 ° grid to obtain cloud top height for comparison to the IASI observations of SO$_2$ plume altitude.

**2.4 UKESM1 simulations**

We conduct a model simulation of the Holuhraun eruption and a corresponding control simulation with no volcanic emissions using the atmosphere-only version 1.0 of the UK Earth System Model (hereafter UKESM1-A) (Sellar et al., 2019; Mulcahy et al., 2020). We compare the perturbation of in-plume cloud properties observed from MODIS to these simulations and we also use the UKESM1-A simulations to further investigate the influence of meteorology on aerosol-cloud interactions.

UKESM1.0 is the first version of the UK Earth System Model and contributed to the sixth Coupled Model Intercomparison Project (CMIP6) (Eyring et al., 2016; Sellar et al., 2019). UKESM1 is based on the HadGEM3-GC3.1 physical climate model (Kuhlbrodt et al., 2018; Williams et al., 2018) coupled to several earth system processes including interactive stratosphere-troposphere chemistry from the UK Chemistry and Aerosol model (UKCA) (Archibald et al., 2020). In the atmosphere-only version of UKESM1 (UKESM1-A), sea surface temperatures and sea ice concentrations are prescribed from the Program for

Climate Model Diagnosis and Intercomparison (Rayner et al., 2003). Vegetation and ocean biological fields are prescribed from a member of the UKESM1 CMIP6 historical ensemble (Sellar et al., 2019).

The aerosol scheme within UKCA is the modal version of the Global Model of Aerosol Processes (GLOMAP-mode) which simulates new particle formation, gas-to-gas particle transfer, aerosol coagulation, cloud processing of aerosol, and deposition

of sulphate, sea salt, black carbon, and particulate organic matter (Mann et al., 2010; Mulcahy et al., 2020). Mineral dust is simulated separately using the CLASSIC dust scheme (Woodward et al., 2001). The aerosol chemistry is coupled to the UKCA

stratospheric-tropospheric aerosol scheme where chemical oxidants are interactively simulated (Archibald et al., 2020). UKCA uses aspects of the Unified Model Global Atmosphere (GA7.1; Walters et al., 2019) within the UKESM for the large-scale advection, convective transport and boundary layer mixing of aerosol. Aerosol particles are activated into cloud droplets using the Abdul-Razzak and Ghan (2000) activation scheme. Large-scale cloud microphysics is a single-moment scheme based on Wilson and Ballard (1999) with improvements based on Boutle et al. (2014). Changes in cloud droplet number concentration ($N_d$) can impact cloud droplet effective radius (Jones et al., 2001) and the autoconversion of cloud liquid water to rain water through the Khairoutdinov and Kogan (2000) scheme. Aerosol–cloud interactions are simulated in large-scale liquid clouds. Convection is parameterized separately to large-scale clouds and does not consider aerosol. Bulk properties of large-scale clouds are simulated using the prognostic cloud fraction and prognostic condensate (PC2) scheme (Wilson et al., 2008a, b) with the modification described in Morcette (2012). The GA7.1 model and its coupling to UKCA are described in further detail in Walters et al. (2019) and Mulcahy et al. (2020).

We use global model simulations with a resolution of N96L85, which is a horizontal resolution of 1.875 x 1.25° (~208 × 139 km at the equator and ~86 x 139 km near the Holuhraun eruption site), with 85 atmospheric levels. The model resolution is coarser than the MODIS and OMPS datasets we use that are at 1.0 x 1.0° resolution. In the Holuhraun eruption simulation of this UKESM1 setup, the volcanic $SO_2$ emissions are distributed equally between 0.8 km and 3 km in the grid cell containing the eruption vent following the magnitude and altitude profile of emissions (Malavelle et al., 2017). The prescribed volcanic $SO_2$ emissions vertical profile is in agreement with satellite observations from the IASI shows the $SO_2$ plume height during September and October 2014 is mostly between 0.8 and 2.5 km (Jordan et al., 2024). We refer to the simulation that includes volcanic emissions as UKESM1-Hol hereafter. A control simulation was also performed without the Holuhraun eruption emissions which we refer to as UKESM1-Ctrl. The control simulation enables us to assess whether any of the differences in our model simulations are simply due to differences in the meteorology, rather than due to the aerosol perturbations. The eruption and control simulations include background aerosol emissions from anthropogenic and natural sources. The modelled horizontal winds between approximately 1.3 to 80 km are nudged towards ERA-Interim reanalysis on a 6-hourly time scale to reduce model internal variability. The model output fields are extracted at high temporal resolution (3 or 6-hourly output) for comparison to observational data. The spatial and chemical evolution of the Holuhraun aerosol pollution in these UKESM1-A simulations has recently been evaluated in a multi-model comparison framework in Jordan et al. (2024).

To aid the comparison of modelled cloud properties with MODIS, we use the COSP MODIS simulator for model output where possible (Bodas-Salcedo et al., 2011, Pincus et al., 2012). $N_d$ was calculated from COSP output using the same calculation and filtering as for the MODIS data. Likewise to the MODIS analysis, we focus on marine liquid clouds. In our plume analysis of the model simulations, we use the OMPS $SO_2$ plume mask that was created from OMPS data regridded to the coarser model resolution.

## 2.5 Trajectory modelling

The Hybrid Single Particle Lagrangian Trajectory (HYSPLIT4) model (Stein et al., 2015) was used to calculate 10-day back trajectories from the Holuhraun eruption vent. For consistency with UKESM1-A simulations, ERA-Interim 6-hourly reanalysis (Dee et al., 2011), re-gridded to 1.0° x 1.0° were used to drive HYSPLIT. For every hour during September 2014, a 27-member ensemble of 10-day backward trajectories was initiated from the eruption site (64.85°N, 16.83°W) at a starting altitude of 2000 m agl (above ground level). The 27-member ensemble was created to sample the uncertainty associated with location accuracy. The centre trajectory of the ensemble is initialised at the coordinates above, with the remaining 26 members offset by a fixed grid factor of 1.0° of latitude/longitude in the horizontal and 0.01 sigma units in the vertical, forming a 3-dimensional space with 27 trajectory initialisation points.

We create transport probability function maps to investigate the dominant movement path of the air masses during September 2014. The transport probability function, $P(A_{i,j})$, represents the probability (%) of a backward trajectory passing through a specific grid cell. $A_{i,j}$ was calculated as:

$$A_{i,j} = \frac{n_{i,j}}{N} \tag{2}$$

Where $n_{i,j}$ corresponds to the number of distinct trajectory visits within a grid cell, and $N$ corresponds to the total number of trajectories. The maps allow a qualitative assessment of whether the air-masses reaching Holuhraun are from geographic areas that are relatively pristine or influenced by anthropogenic emissions and also help characterise the thermodynamic properties of those air-masses.

## 3 Results

### 3.1 Evolution of the Holuhraun SO$_2$ plume

Our analysis uses the plume masks derived from the observed column amount of SO$_2$ to isolate cloud properties inside versus outside the aerosol plume formed from the 2014-15 Holuhraun eruption (see section 2.1). Variability in meteorology and cloud state across a domain can make the impact of aerosol perturbations to cloud properties difficult to isolate, for example, if the aerosol influenced cloud fields experience different conditions than the unperturbed cloud fields (e.g. Christensen et al., 2022). Therefore, we define a bounding box area around our plume mask to minimise differences in meteorological conditions. Figure 1 shows a snapshot of the column amount of SO$_2$ within our plume mask and the corresponding bounding regions for the middle day in each of the four weeks in September 2014 that we analyse. Supplementary Figure S1 shows an animation of the plume mask and bounding region for all the days analysed.

On many of the days in September 2014, the observed SO$_2$ plume disperses to the north-east of the eruption site. There are a handful of days within the month when the plume was transported towards Western Europe where it triggered air pollution events (Ialongo et al., 2015; Schmidt et al., 2015; Boichu et al., 2016; Steensen et al., 2016; Twigg et al., 2016; Zerefos et al., 2017). Our plume masking and bounding box method appears to track the spatial evolution of the observed SO$_2$ plume well for most days in September.

Figure 1 and supplementary Figure S2 show the daily mean total column amount of SO$_2$ for the UKESM1-Hol simulations, and the corresponding plume mask and bounding region when derived from the OMPS-coarse mask. In common with simulations of explosive volcanic eruptions that are nudged to ERA reanalyses (Haywood et al., 2010; Wells et al., 2023), in general the SO$_2$ plume simulated in the model agrees well with the spatial location of the SO$_2$ plume observed from OMPS. Jordan et al., (2024) also show that the UKESM1-Hol simulations accurately capture the evolution of the volcanic plume in September and October 2014 when compared SO$_2$ retrieved from the IASI (Infrared Atmospheric Sounding Interferometer) satellite instrument. This agreement gives us confidence in using the SO$_2$ mask derived from observations to evaluate the model simulations, but there may be days where there are differences in the spatial location of the plume and bounding box derived from observations compared to the model simulations (e.g. 25[th] September). The recommended quality control procedure for OMPS involves excluding pixels where the SZA > 70°. Due to the high latitude of the eruption, this procedure excludes pixels at the top of our domain as September progresses and would also exclude pixels from the MODIS dataset that are less reliable. The column amount of SO$_2$ in UKESM1-Hol therefore has a further northward extent than the OMPS plume mask towards the end of September.

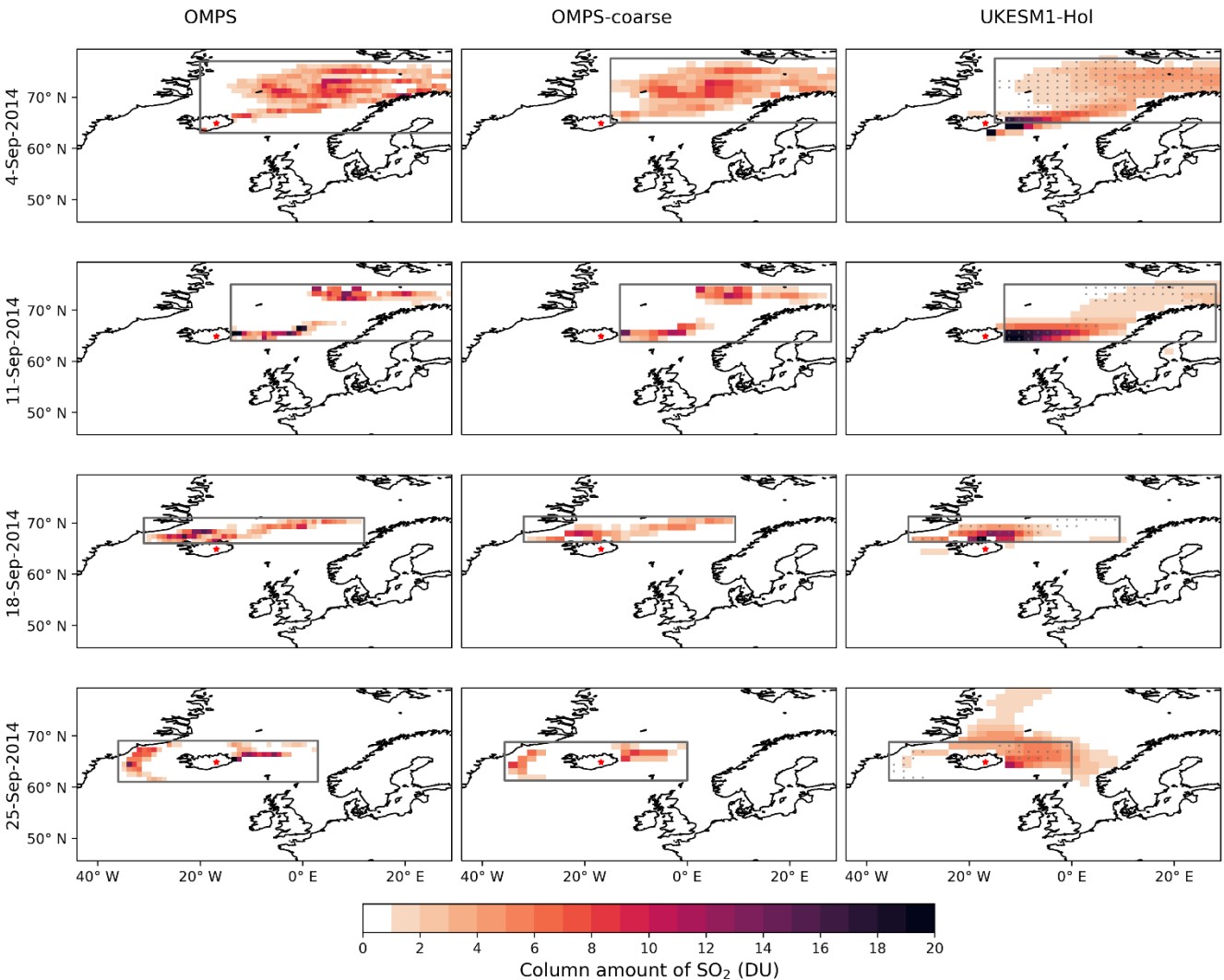

**Figure 1: Total column amount of SO₂ (Dobson Units) retrieved from OMPS (1.0 x 1.0 °), OMPS-coarse (OMPS regridded to UKESM1-Hol resolution) and simulated in UKESM1-Hol within the plume mask for the midweek day of the four weeks in September 2014 being analysed. The plume mask is defined where the total amount of SO₂ exceeds 1 DU. The grey box shows the bounding box region surrounding the plume mask which we conduct our in-plume vs out-of-plume analysis within. The grey dots in the UKESM1-Hol column show the location of the OMPS-coarse plume mask used in the model comparison. The red star shows the location of the eruption site.**



## 3.2 Aerosol perturbation to observed in-plume cloud properties

The next stage of our analysis compares cloud properties retrieved from the MODIS COSP dataset inside the $SO_2$ plume mask to areas outside the plume mask yet still within the bounding region. Supplementary Figure S2 shows the plume mask bounding region overlaid on MODIS observations of marine liquid cloud $N_d$ and $r_{eff}$ for our snapshot days. This figure gives an indication of the spatial variation in cloud properties across the domain as well as the data coverage.

We evaluate if there is an aerosol induced perturbation to $N_d$, $r_{eff}$, LWP and cloud fraction in marine liquid clouds for days in September 2014. Animations of daily cloud properties and their in-plume vs out-of-plume distribution are shown in Supplementary Animations S4-S7. As an example of the daily analysis, Figure S3 shows the distribution of $N_d$ and $r_{eff}$ in-plume and out-of-plume for our snapshot days. For each day we use the Mann-Whitney U test (Mann and Whitney, 1947) to evaluate if the sample of in-plume cloud properties is significantly different to the sample of the out-of-plume cloud properties.
The results of this statistical significance test are summarised in Figure 2.

In more than half the days we analyse (14 out of 24) observed $N_d$ is statistically significantly higher inside the plume compared to outside. Between the 1st to 12th September there are only two days (7th and 10th September) where $N_d$ is not higher within the plume. However, between 14th and 21st September no days display significantly higher $N_d$ inside the plume. If we exclude
14th September due to its small sample size (supplementary Animation S4), the remaining days in this collection fall within the 3rd week of September; which leads us to aggregate our results into the weeks of September later in the study. In the 4th week of September, 5 of the 6 days analysed have significantly higher $N_d$ within the plume. Across our analysis, all but one of the days that display significantly larger values of $N_d$ in-plume have corresponding statistically significantly smaller values of $r_{eff}$ in-plume. An aerosol induced increase in $N_d$ and decrease in $r_{eff}$ is consistent with the Twomey effect (Twomey, 1974)
which has been widely observed (e.g. Christensen et al., 2022). Most days (6 out of 9) within the first two weeks of September that have an increase in in-plume $N_d$ show a significant increase in LWP. One day within the first two weeks show a significant decrease in LWP. Yet the days in the 4th week of September that display an in-plume increase in $N_d$ reveal a different picture with no consistent response in LWP.

To investigate the lack of perturbation to the in-plume $N_d$ for many days of the 3rd week of September and why there is a variation in the in-plume LWP response across September, we aggregate our daily plume analysis into the weeks of September. We also use the weekly-aggregated data to compare the observed in-plume perturbation to cloud properties that are simulated by UKESM1-A. Figure 3 shows the weekly in-plume and out-of-plume distributions for $N_d$ and $r_{eff}$. LWP is shown in Figure 4. The weekly aggregated results confirm our daily plume analysis; there is a statistically significant increase in $N_d$ and decrease
in $r_{eff}$ for the 1st, 2nd and 4th weeks of September, which is absent in the 3rd week. The sample of in-plume LWP is statistically significantly greater in these 3 weeks but not the third week. We next compare our observed weekly plume analysis results to

those from UKESM1-A and use diagnostics available from the model simulations in combination with airmass back trajectory analysis to untangle the differences in the aerosol-perturbation to cloud properties over the first four weeks of the Holuhruan eruption.


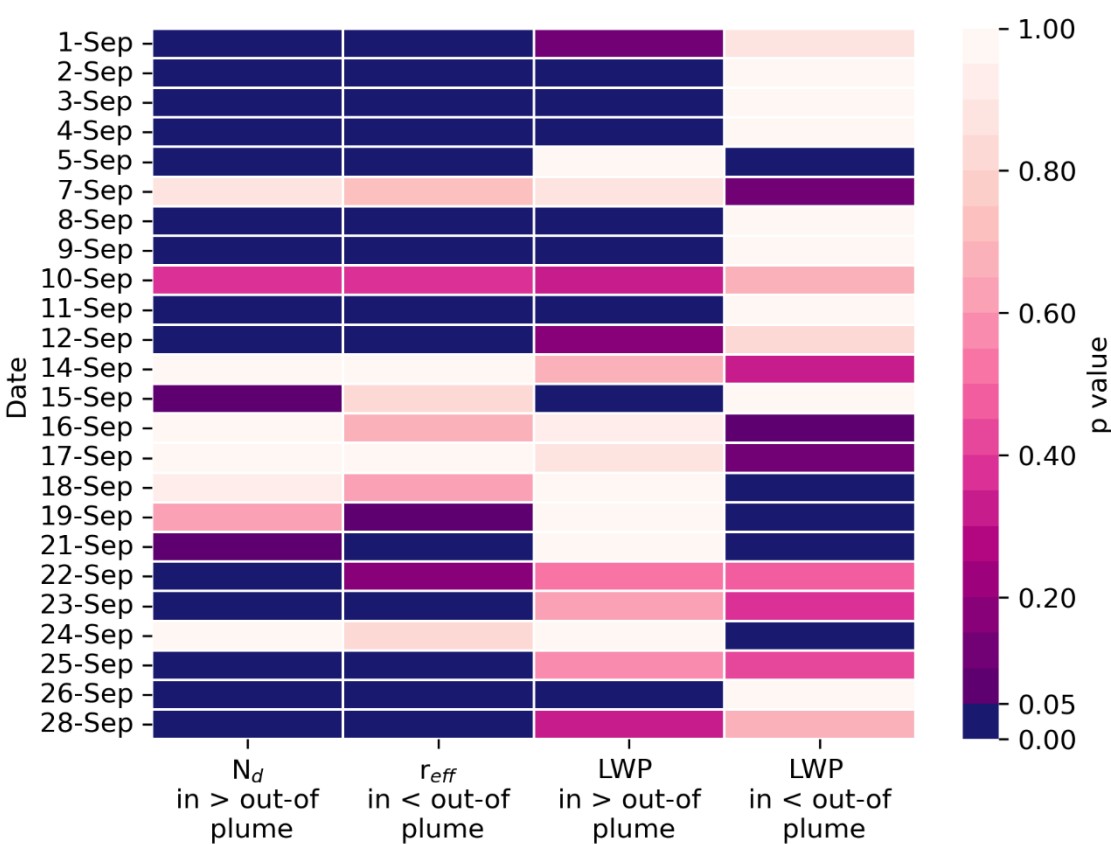

**Figure 2: Statistical significance of daily changes in MODIS marine cloud properties inside vs outside of the SO₂ plume mask. Significance is evaluated using the Mann-Whitney U test. The colour bar displays the p value, with dark blue indicating a statistically significant perturbation to cloud properties inside the plume for that day.**


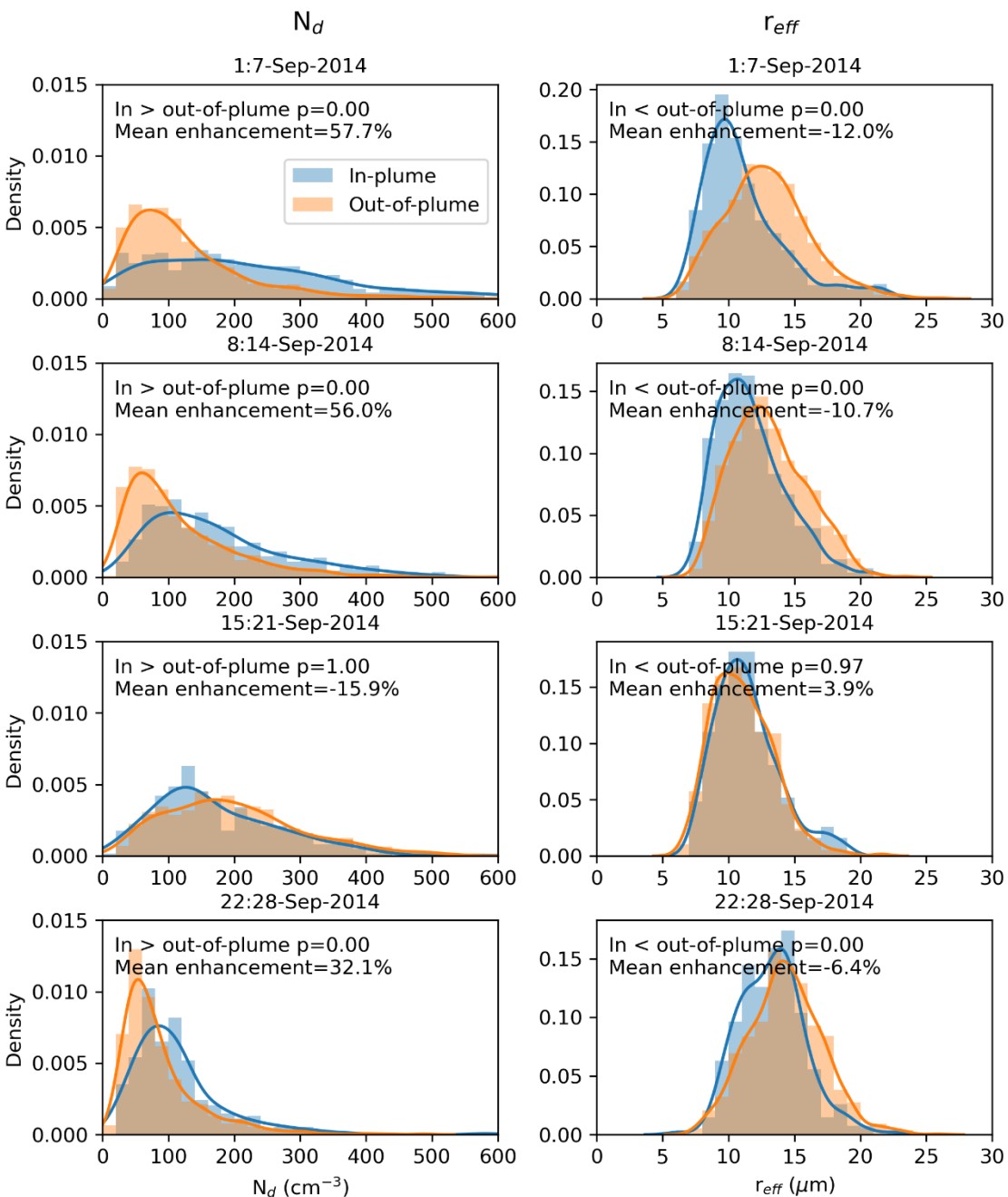

**Figure 3: Histogram of MODIS liquid cloud droplet number concentration (cm$^{-3}$) and effective radius (μm) inside (blue) and outside (orange) the plume mask aggregated by week. Only marine cloud properties are considered. The Mann-Whitney U test is used to calculate if the in-plume $N_d$ is statistically greater than outside of the plume. The p value and mean in-plume enhancement is displayed for each week.**


MODIS LWP

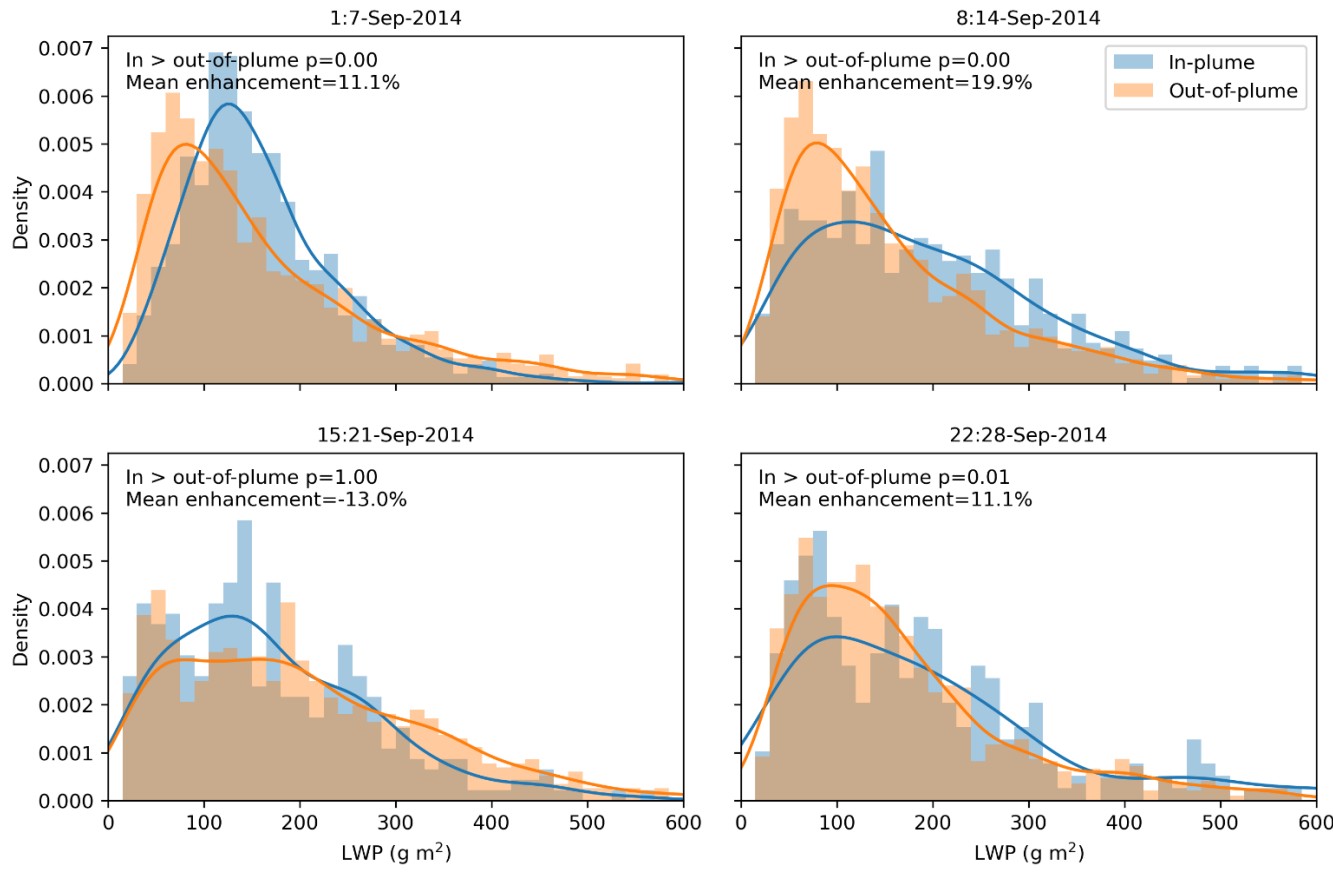

**Figure 4: Histogram of MODIS in-cloud liquid water path (g m$^2$) inside (blue) and outside (orange) the plume mask aggregated by week. Only marine cloud properties are considered. The Mann-Whitney U test is used to calculate if the in-plume LWP is statistically greater than outside of the plume. The p value and mean in-plume enhancement is displayed for each week.**

## 3.3 Comparison of observed vs modelled perturbation to in-plume cloud properties

Table 1 shows the area-weighted geometric mean values of marine liquid cloud properties inside and outside of the plume mask, and the corresponding in-plume perturbation to cloud properties. The UKESM1-Hol simulation shows significantly greater $N_d$ and significantly smaller $r_{eff}$ inside the plume in the first two weeks of September, with no statistically significant perturbation in that direction in the control simulation. The lack of perturbation to $N_d$ and $r_{eff}$ in UKESM1-Ctrl indicates the perturbation to cloud properties inside the plume is not explained by meteorological variability and is therefore aerosol-induced. In the 3rd week, UKESM1-Hol features a significantly decreased $N_d$ inside the plume which is consistent with MODIS and UKESM1-Ctrl. In the 4th week there is a non-significant increase in $N_d$ and decrease in $r_{eff}$ in UKESM1-Hol, in comparison

to a significant change in the MODIS observations. The statistical significance of daily changes in modelled cloud properties is summarised in Figure S5.


There is not a significant increase or decrease in in-plume LWP in UKESM1-Hol during the first two weeks of September. This contrasts with MODIS where the distribution of in-plume LWP values is significantly greater than out-of-plume. In the 3rd week, there is an observed decrease in LWP inside the plume. The in-plume decrease in LWP is represented in the eruption and control simulations, indicating that the decrease in LWP in the 3rd week could be due to the sampling of different cloud

conditions inside the plume rather than an aerosol effect. During the 4th week the distribution of in-plume LWP is statistically greater in MODIS, in contrast to a in-plume reduction in LWP in UKESM1-Hol and UKESM1-Ctrl.

We also evaluate perturbations to cloud fraction in our weekly analysis. In the first week cloud fraction is statistically greater in-plume in MODIS, UKESM1-Hol and UKESM1-Ctrl. The consistency of the increase in in-plume cloud fraction between

UKESM1-Hol and UKESM1-Ctrl indicates that the large in-plume enhancement in cloud fraction is mostly driven from meteorology variability across the field. In the second week, MODIS observations shows a more modest statistically significant increase in cloud fraction in-plume whilst UKESM1-A simulations show a statistically significant decrease in-plume. In the third week in-plume cloud fraction is statistically significantly lower in-plume compared to out-of-plume. Likewise to the first week, the consistency in cloud fraction changes between the UKESM1-Hol and UKESM1-Ctrl indicates this is primarily

driven by meteorological variability. In the fourth week there is a non-significant decrease in observed in-plume cloud fraction, but the decrease is statistically significant in both model simulations. We tested the robustness of our observed cloud fraction results to using different MODIS cloud fraction variables as shown in Figure S6. Total (all phases) cloud retrieval fraction and total cloud mask fraction showed a statistically significant increase in in-plume cloud fraction during the first two weeks and decrease in the third week but the magnitude of response is lowered. Whereas the direction of the response in the 4th week was

of opposite sign to the liquid cloud retrieval fraction.

These results indicate that UKESM1-A captures the observed change in $N_d$ and $r_{eff}$ in the first two weeks of September 2014 but there is not a significant change in simulated in-plume LWP during these two weeks. The model control simulations help elucidate that changes in cloud properties inside the plume during the 3rd week are likely not due to ACI. Next, we use the

UKESM1-Hol simulation and trajectory modelling to investigate the aerosol-cloud interaction mechanisms at play during the different weeks in September 2014.

| | | Inside (outside) plume values | | | | In-plume perturbation (%) | | | |
|---|---|---|---|---|---|---|---|---|---|
| Week | | 1 | 2 | 3 | 4 | 1 | 2 | 3 | 4 |
| $N_d$ (cm$^{-3}$) | MODIS | 148 (94) | 147 (94) | 144 (172) | 95 (72) | **58** | **56** | **-16** | **32** |
| | UKESM1-Hol | 221 (141) | 235 (203) | 188 (188) | 202 (200) | **56** | **16** | **-20** | 8 |
| | UKESM1-Ctrl | 85 (105) | 93 (103) | 101 (130) | 105 (123) | **-20** | -2 | **-24** | **-13** |
| $r_{eff}$ (µm) | MODIS | 11.1 (12.7) | 11.3 (12.7) | 11.9 (12.2) | 13.0 (13.9) | **-12** | **-11** | **4** | **-6** |
| | UKESM1-Hol | 9.6 (11.1) | 9.4 (9.9) | 10.6 (10.0) | 10.2 (10.5) | **-14** | **-5** | **6** | -3 |
| | UKESM1-Ctrl | 12.5 (12.0) | 12.3 (12.3) | 12.0 (11.1) | 12.3 (12.0) | **4** | 0 | **8** | **3** |
| LWP (g m$^{-2}$) | MODIS | 137 (123) | 149 (124) | 134 (154) | 145 (130) | **11** | **20** | **-13** | **11** |
| | UKESM1-Hol | 110 (107) | 110 (116) | 75 (109) | 125 (135) | 3 | -5 | **-31** | -8 |
| | UKESM1-Ctrl | 93 (100) | 85 (92) | 79 (103) | 113 (132) | -7 | -9 | **-30** | **-15** |
| cloud retrieval fraction (%) | MODIS | 66 (36) | 56 (46) | 72 (58) | 28 (33) | **87** | **18** | **-14** | -15 |
| | UKESM1-Hol | 60 (32) | 38 (49) | 24 (44) | 31 (39) | **85** | **-21** | **-45** | **-20** |
| | UKESM1-Ctrl | 55 (32) | 36 (49) | 22 (40) | 32 (37) | **73** | **-26** | **-44** | **-12** |

**Table 1: Weekly area-weighted geometric mean of MODIS and UKESM1-A marine liquid cloud properties inside and outside of the plume mask. The last four columns display the mean in-plume perturbation (%) of each cloud property. The in-plume perturbation is calculated as (mean inside plume- mean outside of plume)/ mean outside of plume where the mean is the area-weighted geometric mean. Bold text in the in-plume perturbations represents where the weekly aggregated in-plume values are respectively statistically greater or less than outside of the plume. Table S1 shows the sample size of $N_d$ weekly aggregated data.**

## 3.4 Disentangling aerosol-cloud interaction mechanisms during September 2014

### 3.4.1 Air mass history

In the previous section we showed that the lack of in-plume perturbation to $N_d$ and $r_{eff}$ in the 3rd week of September featured in both the MODIS observations and the UKESM1-A Holuhraun simulation. In the 3rd week, the MODIS out-of-plume $N_d$ distribution shown in Figure 2 more closely resembles the polluted in-plume distributions of $N_d$ than the clean out-of-plume backgrounds. We use back-trajectory modelling to explore the air mass origins during the different weeks of our analysis. Figure 5 shows that during weeks 1, 2 and 4 back trajectories initialised at the eruption site mostly pass through pristine air to the west of Iceland enroute to the Holuhraun eruption site. However, in week 3, a larger proportion of the back-trajectories pass over Western Europe. The air masses passing over Europe will experience greater aerosol pollution from anthropogenic sources, which is a plausible reason for higher background $N_d$ during week 3. This polluted background is also well simulated by UKESM1-A (Figure S7).

The activation of the Holuhraun aerosol plume into cloud droplet depends on multiple factors. These factors include the number, size and hygroscopicity of aerosol particles, as well as the updraft velocity at cloud base and the water vapour supersaturation. Reutter et al. (2009) showed the activation of aerosol into cloud droplets can occur under three regimes: updraft-limited, aerosol-limited or aerosol- and updraft-sensitive regimes. The updraft-limited activation regime is characterized by low ratios of updraft velocity/aerosol number concertation and hence is more likely to occur under polluted air masses, such as week 3 in our analysis (Jones et al., 1994; Reutter et al., 2009; Carslaw et al., 2013; Spracklen and Rap, 2013). In this updraft-limited regime the activation of aerosol to cloud droplets depends on updraft velocity rather than aerosol concentration. As a result, under this regime, polluted air masses arriving in the region of the Holuhraun aerosol plume during the 3rd week would be less susceptible to further aerosol-induced increases in $N_d$. In comparison, in the aerosol-limited region the activation of aerosol to cloud droplets is proportional to the aerosol number concentration.

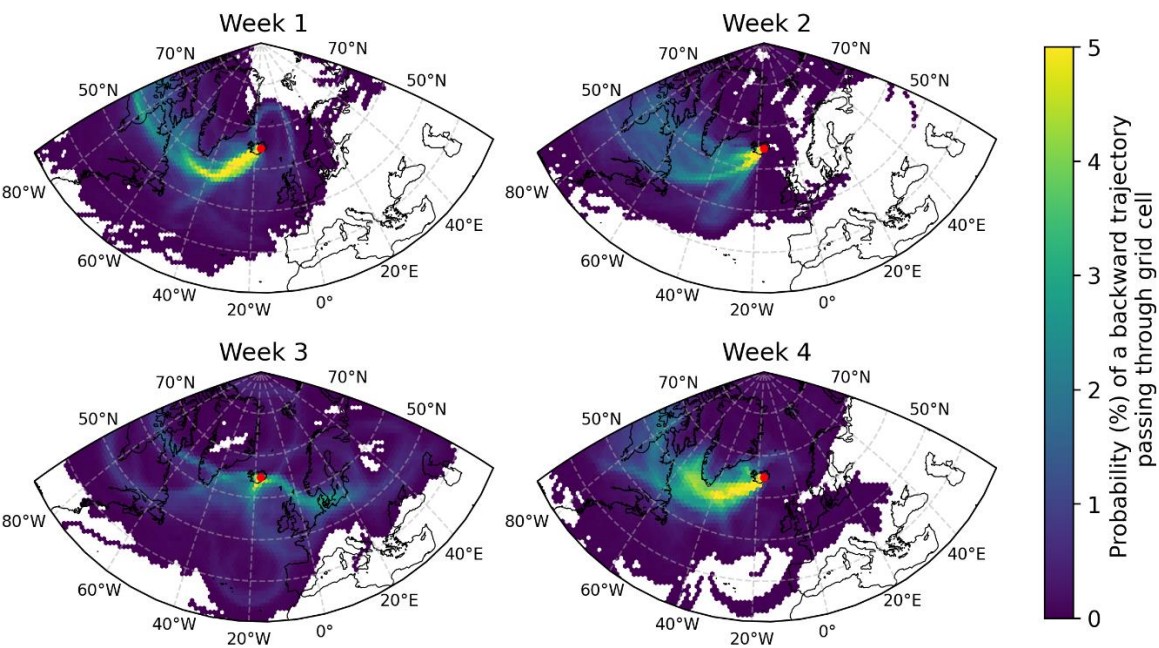

**Figure 5: An ensemble of back trajectories was initialised each hour at 2000 m above the Holuhraun eruption site (64.85°N, 16.83°W), as explained in Section 2.4. The probability (%) of a backward trajectory passing through a specific grid cell ($A_{i,j}$) is shown here. The start dates of the trajectories are grouped by the weeks of our analysis.**

### 3.4.2 Background meteorology

We also explore if the meteorological conditions during the weeks of our analysis affect ACI. During week 3, the MODIS and UKESM1-A simulations in-plume LWP and cloud fraction is lower than outside the plume. In the absence of a clear aerosol-cloud interaction inside the Holuhraun plume, a difference in LWP and cloud fraction may indicate the area inside the plume has different meteorological conditions and cloud properties to outside of the plume. Figure 6 shows visible satellite imagery in the 3rd week overlayed by the plume mask and bounding box region. On 16th – 19th September there is a region of clear sky that persists in the north of the bounding box. Since there is agreement between the lack of ACI signal in observations and simulations in the 3rd week, we use the UKESM1-Hol simulation to investigate differences in meteorological conditions during the 3rd week that may contribute towards the negligible in-plume aerosol perturbation to cloud properties.

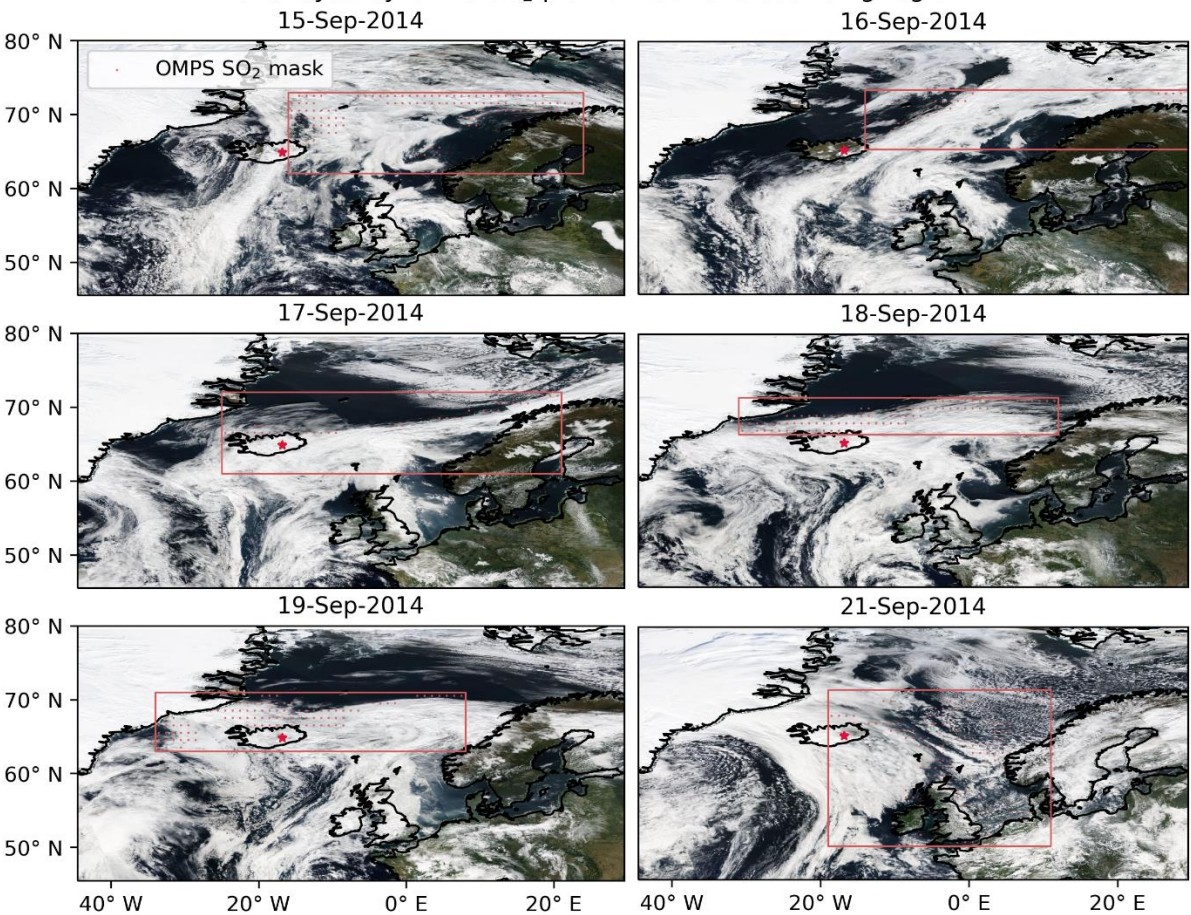

**Figure 6: Visible image from MODIS AQUA for 15th – 21st September 2014. The OMPS SO₂ plume mask and bounding region is overlaid on the visible imagery. 20th September is excluded due to no OMPS SO₂ retrieval on that day. Visible imagery is obtained from the corrected reflectance (true colour) MODIS AQUA data available on NASA Worldview (https://worldview.earthdata.nasa.gov/, last access 1st June 2023).**


Figure 7 shows meteorological variables inside the bounding box in the UKESM1-Hol simulation. The model simulations are nudged to ERA-Interim reanalysis horizontal winds and potential temperatures. The 3rd week is noticeably drier in terms of precipitation and relative humidity at 950 hPa which is representative of the clear-sky region in the north of the bounding box during the 16th – 19th September. There is a slightly lower median and smaller interquartile range of lower tropospheric
stability (LTS) during the 3rd week but there are many outliers that represent grid cells with higher LTS values. The number of outliers with high LTS values implies a contrast in the conditions in the bounding box during 3rd week. A higher LTS value indicates a strong, low-lying inversion that traps moisture more efficiently in the boundary layer and favours greater cloud cover (Wood and Bretherton, 2006). LTS is calcualted as the difference in potential temperature between 720 and 1000 hPa.

Variables affecting the production of sulphate aerosol and the number of aerosols activated to cloud droplets are also shown in Figure 7. The first box plot shows the ratio of vertical mean gas-phase to aqueous-phase production rate of sulphate aerosol ($SO_4^{2-}$) inside the plume. The median and quartiles of the ratio have higher values in week 3. A higher ratio indicates either more gas-phase production or less aqueous-phase production of sulphate which is consistent with the plume location partly covering a region with less cloud during week 3. In the gas-phase, sulphate aerosol is formed through the reaction of $SO_2$ with
OH to form $H_2SO_4$ vapour. Nucleation and condensation then occur to produce aerosols with larger size and number. In UKESM1, these gas-phase aerosol processes produce sulphate aerosol in all size modes whereas in clouds, $SO_2$ dissolves and undergoes oxidation with $H_2O_2$ and $O_3$ to form sulphate (Turnock et al., 2019). The sulphate aerosol produced through in-cloud oxidation is split into the soluble accumulation and coarse modes (Mulcahy et al., 2020). Less aqueous-phase production of sulphate aerosol is therefore in line with the lower values of in-plume soluble accumulation mode aerosol (i.e. an effective
size for droplet nucleation) during week 3. The magnitude of $SO_2$ emissions in the Holuhraun simulations follow that described in Malavelle et al., (2017) (as shown in their Supporting Information). Emissions during the first two weeks of the eruption were larger than during weeks 3 and 4 which also contributes to lower amount of soluble accumulation mode aerosol during these weeks in the Holuhraun simulations. However, emissions were still large at an average estimated as 57.5 kt $SO_2$/day during the latter weeks and we would expect an aerosol perturbation to $N_d$ in an environment susceptible to aerosol
perturbation.

Accumulation mode aerosol dominate the contribution to CCN concentrations over polluted land regions (e.g. Chang et al., 2017). In UKESM1, aerosols are activated into cloud droplets using the activation scheme of Abdul-Razzak and Ghan (2000). Once per timestep the activation scheme calculates $N_d$ at cloud base and imposes it on all grid cells above the cloud base within
the same liquid cloud. The activation scheme also depends on the subgrid vertical velocity variance (West et al., 2014). The box plots show that although soluble accumulation mode aerosol is lower during the last two weeks of September than the first two weeks, the difference in the number of activated particles at the lowest cloud base in the bounding region is less evident. In an updraft-limited activation regime that is more likely to occur under polluted air masses (such as week 3), cloud droplet formation is proportional to updraft velocity and essentially independent of aerosol number concentration (Reutter et al., 2009).

The last two weeks of September exhibit larger variance in subgrid vertical velocity at the lowest cloud base. Hence, an updraft-limited regime would explain why week 3 has a similar number of activated particles at the lowest cloud base compared to other weeks despite lower accumulation mode aerosol inside the bounding box. Haghighatnasab et al. (2022) showed how increasing the updraft velocity can increase the background CCN concentration in the Holuhraun domain in a cloud-resolving model. Yet, further study would be needed to definitively identify the activation regime during each week of our study to

support these results.

        The LWP response to an increase in $N_d$ likely depends on the meteorological conditions present, as noted in the introduction. Our results show a shift in the distribution of MODIS LWP inside the plume during weeks 1, 2 and 4 that results from more values in the range ~ 100-300 g m$^{-2}$ and less values ~ < 100 g m$^{-2}$ inside the plume. An increase in LWP is traditionally

associated with reduced collision coalescence in clouds with smaller droplets that can delay the onset of precipitation and result in the accumulation of in-cloud water content (Pincus and Baker, 1994). LWP has been found to increase in low, precipitating marine liquid clouds below moist air; whereas in thicker, non-precipitating clouds below dry air there may be a decrease in LWP due to an increase in cloud top entrainment (Toll et al., 2019). The simulations show that humid conditions are present during these weeks and some clouds are likely to be precipitating (indicated by reff > 14 µm as shown in Figure 3

and Animation S5) which would be in support of conditions favourable for an increase LWP. However, the in-plume LWP in the Holuhraun simulation were not significantly greater than the values out-of-plume during weeks 1, 2 and 4 which contrasts with climate models' tendency to produce unrealistic large increase in LWP when $N_d$ increases (Malavelle et al., 2017, Toll et al., 2019). A weak LWP response to aerosol perturbation in UKESM1-Hol is consistent with results from HadGEM3-UKCA that is an earlier version of the aerosol-climate model used in this work (Ghan et al., 2016, Zhang et al., 2016, Malavelle et al.,

2017). Ghan et al. (2016) hypothesized that the weak LWP respomse in HadGEM3-UKCA could be partly due to the autoconversion scheme used. If instead the meteorological conditions were favourable to the entrainment processes that can decrease LWP, we would not expect a decrease in LWP to be simulated since most current and previous generations of climate models do not include a parameterization where aerosol can impact cloud top entrainment (Toll et al., 2019). We do not discuss the LWP response in week 3 further here due to the missing causal processes of ACI.


        3.4.3 Limitations of using observed column amount of SO$_2$ to identify the aerosol plume
        In our analyses we use the column amount of SO$_2$ to track the aerosol plume as this information is readily available from satellite observations and model simulations. We assume that the column amount of SO$_2$ is a good proxy for where sulphate aerosol is produced, as this information is not observable from satellite observations. Figure S1 shows how the column amount

of SO$_2$ compares to the vertical mean sulphate mass concentration in the UKESM1-Hol simulations. The spatial location of sulphate aerosol is in good agreement with the location of the column amount of SO$_2$ for our snapshot days. However, the unmasked sulphate mass concentration is elevated across a larger area both inside and outside of the plume mask bonding box. The more widespread enhanced aerosol load revealed by the sulphate mass concentration, in combination with slight

differences between the modelled and observed $SO_2$, is likely why the out-of-plume $N_d$ in the UKESM1-Hol concentration is larger than in the UKESM1-Ctrl. The absolute values of $N_d$ observed by MODIS are lower than in UKESM1-Hol, and the MODIS out-of-plume $N_d$ is comparable to the out-of-plume $N_d$ in UKESM1-Ctrl.

In addition, the OMPS column amount of $SO_2$ does not provide information on when the sulphate plume is within the cloud layer where the aerosol-cloud perturbation takes place. Therefore, we compare the $SO_2$ plume height obtained from IASI (Carboni et al., 2016) and the height of the maximum $SO_2$ mole fraction form the UKESM1-Hol simulations to the liquid cloud height obtained from MODIS Aqua. Animation S8 shows for each day when the $SO_2$ plume height in the IASI observations and the height of maximum $SO_2$ mole fraction in the Holuhraun simulations is below or above the MODIS Aqua liquid cloud top. On most days, there are grid cells within the $SO_2$ plume that are both above and below the observed liquid cloud top height. Animation S9 shows the vertical mean profile of UKESM1-Hol $SO_2$ mole fraction, IASI $SO_2$ plume height and MODIS liquid cloud top height when averaged over latitude. The MODIS Aqua liquid cloud top height in the latitudinal mean is close to the altitude of maximum $SO_2$, with $SO_2$ generally spanning above and below this height. Therefore, we expect the sulphate aerosol produced in the $SO_2$ plume to be interacting with liquid water clouds. However, isolating when the sulphate aerosol plume is interaction with clouds is difficult to decipher and a limitation of using satellite observations alone.

The analysis of sulphate mass aerosol and $SO_2$ plume height shows the limitations in identifying an aerosol plume mask for informing satellite-model comparisons. At smaller scales, the near-infrared reflectance observed by MODIS Terra has been used to identify polluted clouds and unpolluted clouds (Trofimov et al., 2020).

510

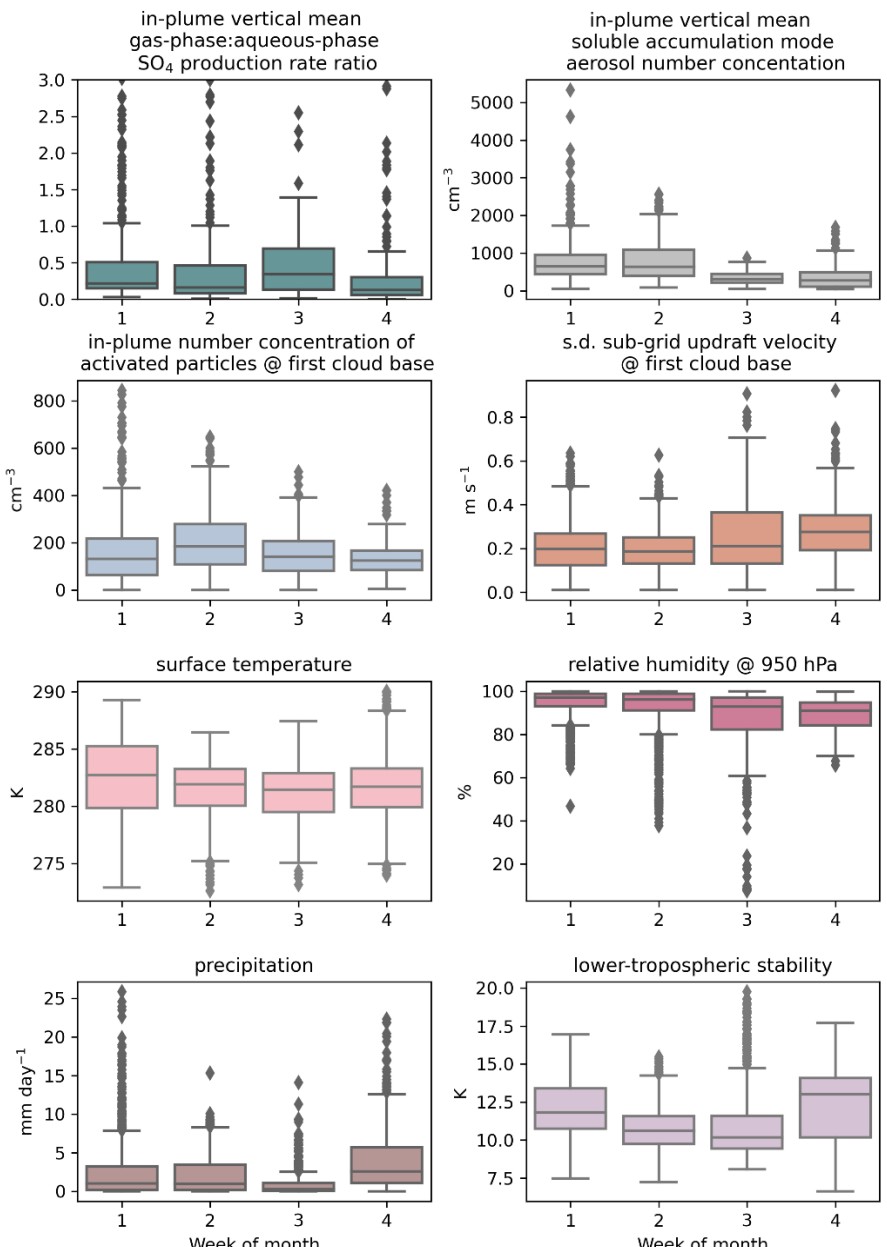

**Figure 7: Box plots of UKESM1-Hol meteorological variables within the OMPS plume mask bounding box. The variables shown are ratio of vertical mean gas-phase to aqueous phase production rate of $SO_4$, vertical mean soluble accumulation mode aerosol number concentration, number concentration of activated particles at first cloud base, standard deviation of sub-grid updraft velocity at first cloud base, surface temperature, precipitation, relative humidity at 950 hPa and lower-tropospheric stability. The daily mean data within the bounding box are aggregated into the four weeks. The first 3 box plots show the in-plume values. The y axis of the $SO_4$ production rate ratio was adjusted to show the box as there was outliers with high values. The box plots show the interquartile range and the median, with the whiskers denoting 1.5 times the interquartile range, and outliers that are defined as outside this range shown as diamond points.**

**4 Discussion and conclusions**

Opportunistic experiments with a known aerosol source, such as degassing volcanic eruptions, offer a way to investigate aerosol-cloud interactions (e.g. Christensen et al., 2022). Our study has built on previous analyses of ACI following the 2014-15 Holuhraun eruption (McCoy and Hartmann, 2015; Malavelle et al., 2017; Chen et al., 2022; Haghighatnasab et al., 2022). We utilise an in-plume versus out-of-plume analysis approach to isolate aerosol perturbations to marine cloud properties in satellite observations and UKESM1-A simulations, and trajectory modelling to understand the impact of airmass history on ACI. Particularly we build on the study of Haghighatnasab et al. 2022 who also used a plume analysis approach, but we use a more detailed plume tracking method and extend the plume analysis approach to the rest of September. The extension of the analysis time frame allows us to group our analysis into weeks that experience differing airmass history and meteorological conditions and elucidate their role on ACI.

We have shown during the first two weeks of September that there is an increase in $N_d$ and decrease in $r_{eff}$, observed, and simulated by UKESM1-Hol when the eruption aerosol plume likely interacts with liquid clouds. As expected, the increased $N_d$ and decreased $r_{eff}$ inside the plume are not reproduced in UKESM1-Ctrl, indicating the perturbation is due to ACI and not differences in meteorology. Our results, which reveal an increase in $N_d$ and decrease in $r_{eff}$ due to Holuhraun eruption aerosol plume are in line with previous ACI studies of the eruption (McCoy and Hartmann, 2015; Malavelle et al., 2017; Haghighatnasab et al., 2022; Chen et al., 2022). However, during the 3rd week in September an increase in $N_d$ is neither observed nor modelled. In the 4th week of September, we observe an increase in $N_d$ and decrease in $r_{eff}$, but an insignificant change in the simulations. To understand what caused the different responses of clouds to increased aerosol across the weeks of our analysis, we used trajectory modelling to track the air mass history in the region, alongside assessing the meteorology and activation of aerosols into cloud droplets using the UKESM1-A simulations.

The 10-day back trajectories reveal that air masses arriving at the Holuhraun eruption site during the 3rd week will likely be more polluted than the other weeks due to passing over Western Europe rather than originating in pristine regions. Polluted air masses are also more likely to experience updraft-limited rather than aerosol-limited activation into cloud droplets (Reutter et al., 2009). Hence, the conditions in the 3rd week may be less susceptible to further aerosol-induced increases in $N_d$ than the other weeks of our analysis due to the polluted background (e.g. Jones et al., 1994; Carslaw et al., 2013). The meteorological fields in the UKESM1-Hol simulation show the 3rd week is drier in terms of relative humidity and precipitation, with the satellite imagery indicating a region of persistent clear-sky in the north of the bounding box region the likely cause. The meteorological conditions during the 3rd week therefore support the higher ratio of gas-phase to in-cloud production of sulphate aerosol which produces less soluble accumulation mode aerosol in the 3rd week, the dominate aerosol mode in the contribution to CCN concentrations over polluted land regions. Overall, we therefore conclude that a combination of the airmass history and background meteorological factors strongly influence aerosol-cloud interactions in the third week. The

ability of background $N_d$ and meteorology in the modulation of ACI, illustrates the importance of improving knowledge of background conditions for accurately calculating ACI. For example, the pre-industrial aerosol loading is a dominant source of uncertainty in present-day aerosol ERF (Carslaw et al., 2013), and present-day analogues to pristine environments can contribute towards constraining aerosol forcing uncertainty (McCoy et al., 2020b; Regayre et al., 2020).

We assessed the LWP response in the three weeks where we isolated an observed shift to smaller and more numerous liquid cloud droplets inside the aerosol plume. We find an observed decrease in the likelihood of small LWP values ($< \sim$100 g m$^{-2}$) and increase in likelihood of LWP values in the range of $\sim$100-300 g m$^{-2}$ inside the plume, resulting in a statistically significant increase in in-plume perturbation LWP. While Malavelle et al. (2017) and Chen et al. (2022) did not isolate an observed perturbation to LWP in monthly means, Haghighatnasab et al., (2022) showed an in-plume decrease in the probability of values

with low LWP and an increase of values with high LWP in satellite observation and cloud-resolving simulations for the 1st week, which is consistent with our results. Cloud-resolving simulations of the Holuhraun eruption suggest there is a decrease in light rain and increase in heavy rain during the 1$^{st}$ week (Haghighatnasab et al., 2022). A decrease in light rain may be due to reduced collision coalescence of smaller droplets that can delay precipitation, and lead to droplets growing larger in size before precipitating, increasing heavy rain and shifting the distribution of in-plume LWP values (Fan et al., 2016;

Haghighatnasab et al., 2022). This mechanism of an increase in LWP due to precipitation suppression supports our observed increase in LWP values inside the plume during the first two weeks of September. However, in UKESM1-Hol, the distribution of LWP values in-plume is not significantly different to out-of-plume. Malavelle et al. 2017 showed that HadGEM3-UKCA (a previous generation of the aerosol-climate model used in UKESM1) produced a minimal LWP response following the Holuhraun eruption, but that models generally overestimate the increase in LWP due to increased aerosol (Malavelle et al.,

2017; Toll et al., 2017).

Chen et al. (2022) showed a significant increase in satellite observations cloud fraction following the Holuhraun eruption when using a machine learning approach that accounts for meteorological confounders. Consistently, our results show an observed increase in cloud fraction during the first two weeks of September 2014. In the first week the increase is simulated by the

volcanic and control UKESM1 simulations, although the increase in cloud fraction is larger in the volcanic simulation. However, in the second week the simulations show a decrease in cloud fraction. In the fourth week, there is a non-significant decrease in observed cloud fraction but a significant decrease in the model simulations. The similarity in the in-plume perturbation to cloud fraction between the volcanic and control simulations across our analysis indicates much of the simulated cloud fraction change is likely dominated by meteorological covariability. Further simulations would be needed to isolate if

the smaller differences between the in-plume perturbation to cloud fraction in the control and Holuhraun simulations could be attributed to aerosols. For example, Grosvenor and Carslaw (2020) examined the contributions of changes in $N_d$, LWP and cloud fractions to pre-industrial to present-day aerosol ERF in UKESM1-A. Their results showed that LWP and cloud fraction were the dominant terms in the radiative forcing of aerosol-cloud interactions over the North Atlantic, and that cloud fraction

changes are more dominant in regions of broken cloud. An additional simulation was conducted in the Grosvenor and Carslaw

(2020) study where $N_d$ was prevented from modifying rain formation through the autoconversion parameterisation, and in these

simulations there was a negligible change in cloud fraction over the North Atlantic.

To conclude, the causal chain of events highlighted over two decades ago (e.g. Haywood and Boucher, 2000) of increases in cloud droplet number concentration decreasing cloud effective radius (Twomey, 1974), which delays auto-conversion and

precipitation processes leading to greater cloud liquid water (Albrecht, 1989) appears to apply in this study. We recommend that ensembles of climate model simulations (e.g. Jordan et al. 2024), higher resolution nested simulations and a more comprehensive use of a Lagrangian framework (e.g. Coopman et al., 2018,) of this opportunistic experiment would provide a more detailed assessment on the causality of meteorological conditions affecting the aerosol perturbation to cloud properties.


**Code and data availability**

The MODIS cloud products from the MODIS COSP dataset (MCD06COSP_D3_MODIS) and from Aqua (MYD08_L2) used in this study are available from the Atmosphere Archive and Distribution System Distributed Active Archive Center of National Aeronautics and Space Administration (LAADS-DAAC, NASA), https://ladsweb.modaps.eosdis.nasa.gov. The OMPS $SO_2$ (OMPS_NPP_NMSO2_PLC_L2 v2) data used in this study is available to download from GES-DISC, NASA, https://disc.gsfc.nasa.gov/datasets/OMPS_NPP_NMSO2_PCA_L2_2/summary. Simplified data and code required to reproduce the main figures in this article are provided on Zonodo (https://doi.org/10.5281/zenodo.12664099; Peace et al. 2024). All other underlying datasets generated and/or analysed during the current study are available from the corresponding author upon reasonable request.

**Author contributions**

AP and JH designed the study. Gridded MODIS Aqua data was created from Level 2 products by YC. GJ ran the model simulations. ED and DP provided guidance in the use of HYSPLIT trajectories. Data analysis and figure preparation was completed by AP. All co-authors provided discussion on the interpretation of results. AP wrote the manuscript with advice from all co-authors.

**Acknowledgments**

JH, DP, YC, AP, and ED would like to acknowledge funding from the NERC ADVANCE grant (NE/S015671/1). GJ and JH were funded under the European Union's Horizon 2020 research and innovation programme under the CONSTRAIN grant agreement 820829. GJ, JH and FM are supported by the Met Office Hadley Centre Climate Programme funded by BEIS. DP would like to express his gratitude to Zak Kipling for providing support in obtaining HYSPLIT input files from ERA-Interim reanalysis data. We would like to thank Paul Kim who helped develop the running and plotting framework for HYSPLIT trajectories and Andy Jones for helping with the experimental set up of UKESM1. We acknowledge the use of imagery from the NASA Worldview application (https://worldview.earthdata.nasa.gov), part of the NASA Earth Observing System Data and Information System (EOSDIS).

**Competing interests**

The authors declare that they have no conflicts of interest.

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
