# Peer review of "In-plume and out-of-plume analysis of aerosol-cloud interactions derived from the 2014-15 Holuhraun volcanic eruption"

_EGUsphere, 2024_

## Referee Comment (RC1)

**Review of "In-plume and out-of-plume analysis of aerosol-cloud interactions derived from the 2014-15 Holuhraun volcanic eruption" by Amy H. Peace et al.**

This manuscript assesses the impact of the 2014-15 Holuhraun volcanic eruption on cloud properties over the region for the month of September and evaluated the influence of varying meteorological conditions at synoptic scale on aerosol-cloud interactions. The authors took a plume analysis approach, in which the aerosol effect is evaluated as the contrast between in- and out-of- plume, using both satellite observations and UKESM1 simulations. This approach has been previously used to study ACI associated with the 2014-15 Holuhraun event, but the authors stated that they upgraded the plume-masking method and extended the analysis time period. They found statistically significant cloud microphysical changes (increase in $N_d$ and decrease in $r_{eff}$) during the first two weeks, both in observations and in simulations. A statistically significant shift in LWP distribution is observed but not modelled in the first two weeks. A synoptic driven shift in airmass origin is observed in week-3 and attributed to the lack of observation and simulation of aerosol effects during week-3.

The manuscript is well written and easy to follow in general, but I did find some places where inherit assumptions and justifications need to be clarified. This work is intriguing in many ways and the topic is of great interest to others in the community, but I think the authors should consider/address these major points (listed below) first to make the conclusions publishable.

**Major comments:**

- A major question I have after reading this work is that where is the cloud relative to the volcanic plume, is the vertical distribution of $SO_2$ (and its relative location to the cloud layer) changing with time, and how does this affect the cloud property changes you observed/modelled and the interpretation of them? To be more specific…
    o What is eruption (injection) altitude? And what is the typical cloud top height in this region?
    o Are plumes always in contact with the clouds? Or there are times when they are separated?
    o How good is your assumption of equally distributed $SO_2$ between 0.8 to 3 km in your ESM setup?
- Regarding the results from UKESM simulations:
    o When interpreting/comparing modelled results to observed results, how do you address the fact that LWP adjustment due to aerosol perturbations is uni-directional by design in ESMs (i.e., only precip-suppression is parameterized, and entrainment-feedbacks are not represented), meaning modelled increase in LWP is likely exaggerated.
    o What, physically, can we learn from comparing LWP in- and out-of- plume in the UKESM-Hol simulation when we know it's not representing the full chain of LWP responses?
    o Have the authors considered running ensemble simulations to relax the assumption on uniformly distributed $SO_2$ profile by varying it to create ensemble members?
    o When I compare cloud properties in-plume between Hol and Ctrl simulations, I see much stronger changes (which we know is causal aerosol effect) compared to the results you get from the in- vs out- plume method; how do you reconcile this

difference? Does this mean the in vs out method is still heavily confounded by meteorological covariations?

o What's the interpretation of the large difference in out-of-plume cloud properties between -Hol and -Ctrl? Aerosol effects or meteorological difference between simulations?

- I wonder as the eruption goes on, do you see a dilution effect of the $SO_2$ plume, such that the background (out-of-plume) is getting more polluted with time? Does this contribute to you week-3 lack of signal in $N_d$? I wonder if you could show histograms of $SO_2$ comparing in- and out-of- plume, similarly to Fig. 3?

- Your bounding box size varies from day-to-day, meaning the degree of meteorological confounding effect also varies from day-to-day in your analysis, how do you address this issue when you compare cloud changes among days and group them into weeks? Also, since some boxes cover land, do you screen out land clouds? I think you only mention this in the caption of one of the figures, I would bring it up clearly in the methodology.

- I had hard time wrapping my head around the LWP responses, particularly when the statistical testing method disagrees with the mean changes between in- and out-. I understand the source of discrepancy, but I am a bit concerned about a lack of high-level take-away of these results in the context of ACI, i.e., should we take away with the message that LWP response is weak and its sign is undiscernible? Personally, I prefer the statistical testing method that focuses on the distribution shift rather than a change in the mean, which could be driven by outliners and not representing a physical response.

- Regarding the satellite retrieved cloud properties,
    o Weren't you concerned about getting mixed-phase or ice clouds when your 1-5km cloud-top constraint is well above the freezing level?
    o For $N_d$ retrievals, high-SZA and low-CF have been shown to produce unreliable retrievals (e.g., Grosvenor et al. 2018), I would put extra constraints on these two variables.
    o Does $SO_2$ plume in the scene affect satellite microphysical (tau and $r_{eff}$) retrievals?

**Minor comments:**

▪ What's the main reason and benefit for extending analysis to 4 weeks? Are you targeted to capture temporal evolution in ACI and/or its timescale? Or just want to explore the influence of different synoptic/meteorological patterns? If it's the latter, why only one month is analyzed when you can do this for the whole eruption time period? Why group your analysis by week (is there any physical reason? Timescale assumption? Any major synoptic pattern shift at weekly scale)? I think these inherit assumptions and justifications need to be layout upfront clearly.

▪ Information on UKESM initialization and boundary conditions needs to be added.

▪ Need to mention why there are missing days in the analysis (e.g., Fig. 4).

▪ Line 87, check sentence "… to and …"

▪ Figures, missing lat/lon labels when maps are shown, and please add color code to the caption.

▪ Line 213, "e.g." should be inside the parenthesis.

▪ Line 226, I think Sep-11 and Sep-25 in Fig. 2 are examples where they do not agree, please discuss and reword.

- Line 231, you meant "north" of the domain? as "top" refers to the vertical direction.
- Line 270-271, after reading through the whole manuscript, I am still missing an explanation on week-4 LWP responses, observed and modelled.
- I feel Fig. S2 is worthy of being included as a main figure.
- Line 303, what is the "reason" that you're referring to? And how do you know it's necessarily the same reason in the simulation, rather than different reasons leading to the same results? perhaps need to reconstruct this sentence.
- Table 1, cloud fraction responses in simulations? Line 328, perhaps better to define perturbation in the main text and use a mathematical expression.
- Line 344-347, this discussion in not very clear and hard to follow, especially the use of terms like 'updraft-limited' and 'aerosol-limited', which are not straightforward concepts and need to be explained/introduced.
- Line 365-371, why choose modelled meteorological conditions instead of ERA reanalysis? And define how is LTS calculated.
- Lines 412-414, any speculation on why this is the case?
- Lines 484-486, I think this needs to be mentioned upfront, I have been wondering about CF responses when I read the results.

**References**

Grosvenor, D. P., Sourdeval, O., Zuidema, P., Ackerman, A., Alexandrov, M. D., Bennartz, R., Boers, R., Cairns, B., Chiu, J. C., Christensen, M., Deneke, H., Diamond, M., Feingold, G., Fridlind, A., HÃŒnerbein, A., Knist, C., Kollias, P., Marshak, A., McCoy, D., Merk, D., Painemal, D., Rausch, J., Rosen- feld, D., Russchenberg, H., Seifert, P., Sinclair, K., Stier, P., van Diedenhoven, B., Wendisch, M., Werner, F., Wood, R., Zhang, Z., and Quaas, J.: Remote Sensing of Droplet Number Con- centration in Warm Clouds: A Review of the Current State of Knowledge and Perspectives, Rev. Geophys., 56, 409–453, https://doi.org/10.1029/2017RG000593, 2018.

---

## Referee Comment (RC2)

Review of ''In-plume and out-of-plume analysis of aerosol-cloud interactions derived from the 2014-15 Holuhraun volcanic eruption'' by Peace et al., submitted to Atmospheric Chemistry and Physics (ACP)

Manuscript number: "acp-2024-360"

Decision: "Major revision"

The study focuses on the uncertainty surrounding aerosol effective radiative forcing (ERF) and aerosol-cloud interactions, which are crucial for understanding climate sensitivity and predicting future climate change. Using the 2014-15 Holuhraun volcanic eruption as a natural experiment, the researchers evaluate the impact of volcanic aerosols on cloud properties during the first month of the eruption, comparing observations with simulations from the UK Earth System Model (UKESM1-A). During the initial two weeks of the eruption, both observations and simulations show a shift to smaller and more numerous cloud droplets within the volcanic plume, along with changes in liquid water path (LWP) values. However, in the third week, this shift is neither observed nor accurately modeled, and discrepancies exist between observations and simulations in the fourth week. The study underscores the influence of air mass history and background meteorological factors on aerosol-cloud interactions across different weeks. Most parts of the manuscript are well written, but there are several issues to be addressed. Based on the descriptions outlined above, my decision is "major revision," and I encourage the authors to revise the manuscript.

**Major comments:**

Definition of Volcanic Plume:

Using $SO_2$ as a proxy for the volcanic plume in the 3rd and 4th weeks following an eruption may not be entirely accurate, as the $SO_4$ formed from it can dissipate from the source and the location of the $SO_2$ plume and $SO_4$ plume could be different. Therefore, it is possible that in the bounding boxes outside the plume area, some fingerprints of $SO_4$ could still be present. Have you thought about comparing the $SO_4$ plume with the $SO_2$ plume in your volcano simulations? I recommend providing a figure similar to Figure 2 for UKESM1-A, but with plots for $SO_4$. Or alternatively provide a comparison in distribution of one of the variables such as cloud droplet effective radius for outside of plume in simulations with the volcano eruption and without the volcano eruption.

UKESM1-A simulations:

I believe that additional details regarding your simulations are necessary. Could you please provide information about the cloud microphysics, cloud cover, and convection scheme utilized in your simulations?

- Additionally, since you use the satellite simulator from the COSP package, were subcolumns employed, and if so, how many subcolumns were utilized given the coarse resolution which is used?

- I'd like to inquire whether the information of size distribution of hydrometeors used as an input for the MODIS simulator, which is essential for simulating MODIS signals, was taken into account in your simulations?
- Regarding the resolution of your simulations, which is mentioned as 1.875 x 1.25° at the equator. What does this resolution correspond to in the North Atlantic, where your analysis takes place? Additionally, it's noted that OMPS and MODIS data are gridded at 0.5 x 0.5-degree resolution. Could you please explain how this resolution compares to your simulation's resolution?

The LWP response:

- In your paper, there is significant discussion regarding the LWP response. I believe it is essential to include analyses of LWP for different weeks of the study in the main manuscript. Therefore, I suggest providing a figure similar to Figure 5 for LWP.
- Does your analysis of precipitation in Figure 8 for the first two weeks of the eruption indicate any suppression in precipitation?

Discussion about cloud fraction:

- Could you elaborate on why analyses in Table 1 for cloud fraction are not included for simulations?
- It would improve the discussion on cloud fraction to compare the results obtained from MODIS with those presented in the study by Chen et al. (2022), which suggests enhancing cloud fraction appears to be the leading cause of climate forcing.

**Minor comments:**

- In Table 1, can you please explain why the mean value for $N_d$ for outside plume in control and Hol simulations for week 3 and 4 is very different? I believe this also can be related to my first comment on the definition of volcano plume.
- Do the simulations conducted with UKESM1-A cover the entire globe? It would be beneficial to briefly discuss the advantages and disadvantages of employing global simulations compared to regional cloud-resolving simulations.
- Please provide some short information in the manuscript regarding the time at which you analyzed simulations. Is it the daily mean or at the time of MODIS-AQUA overpass?
- In Figure 5, regarding the average enhancement observed for $r_{eff}$ and $N_d$, I'm curious about why the decrease in mean enhancement for $r_{eff}$ is more pronounced in weeks 4 and 2 compared to week 1, while the increase in mean enhancement in $N_d$ is less pronounced in weeks 2 and 4 compared to week 1. Regarding the fact that in the relationship used to calculate $N_d$, it seems that $r_{eff}$ has a stronger impact compared to cloud optical depth.
- I propose moving Figure 3 to the supplementary materials or combining the information in the legend with Figure 5.
- In line 191, did you mean event by vent?
- In Figure 4 for $r_{eff}$, it is demonstrated in< out the plume while in figure 5, for $r_{eff}$, the in> out plume is demonstrated. I recommend maintaining consistency between these two figures.

- In the abstract, can you discuss briefly how LWP has changed in the first 2 weeks? In the current version, it is just mentioned that it is changed.
- In Figure 5 caption, Nd should be $N_d$.

---

## Author Comment (AC1)

**Response**

We would like to thank the reviewers for their time and constructive feedback on the manuscript. In particular, following the reviewer suggestions, we have added the analysis of modelled cloud fractions. Below is a summary of the major changes we made following the reviewer suggestions, with more detailed response to individual comments in blue on the following pages.

Major changes:

- One of the main comments was that cloud fraction for the UKESM1-A simulations was missing from Table 1. The values for MODIS cloud fraction in Table 1 originated from MODIS AQUA Level 2 cloud mask fraction. When undertaking this additional analysis, we realised that the cloud mask fraction values were not consistent with values in recent literature (e.g. Haghighatnasab et al. 2022 and Chen et al. 2022) and with our model output. We tested different variables of cloud fraction from the recently released MODIS COSP L3 dataset and found the variable 'liquid cloud retrieval fraction' to be more consistent. The MODIS COSP dataset was not available when the work for this paper started, but we now feel using this L3 dataset is the most robust way to compare observed and simulated cloud properties. We have therefore updated our analysis throughout the paper to use the MODIS COSP L3 dataset. The main conclusions of our work do not change with the new dataset.
- Previously we were using MODIS AQUA L2 products gridded to OMPS resolution (0.5 x 0.5 degrees). The plume mask was created using the 0.5 degree resolution OMPS data. The MODIS COSP dataset is 1 x 1 degrees, so now we regrid OMPS swaths to the 1 x 1 degree resolution and calculate the plume mask. We found that at this courser resolution it is not necessary to apply the median filtering to reduce individual pixels where the column amount of SO2 > 1 DU.
- For LWP our previous results showed that there could be a statistically significant shift of in-plume LWP values to higher LWP but a negative in-plume mean enhancement. Reviewer 1 mentioned that taking a high-level conclusion from these contrasting results was difficult. In response, we have changed the method for calculating the in-plume enhancement. We previously took the regional mean of in-plume and out-of-plume values for each day and then calculated the arithmetic mean over the week. Instead, we now calculate the geometric mean for the week-aggregated values using area weights, which is more appropriate for skewed distributions. When using the geometric mean method, the direction of the in-plume mean enhancement is consistent with the statistical testing of the distribution across our results.

**Review 1**

**Review of "In-plume and out-of-plume analysis of aerosol-cloud interactions derived from the 2014-15 Holuhraun volcanic eruption" by Amy H. Peace et al.**

This manuscript assesses the impact of the 2014-15 Holuhraun volcanic eruption on cloud properties over the region for the month of September and evaluated the influence of varying meteorological conditions at synoptic scale on aerosol-cloud interactions. The authors took a plume analysis approach, in which the aerosol effect is evaluated as the contrast between in- and out-of- plume, using both satellite observations and UKESM1 simulations. This approach has been previously used to study ACI associated with the 2014-15 Holuhraun event, but the authors stated that they upgraded the plume-masking method and extended the analysis time period. They found statistically significant cloud microphysical changes (increase in Nd and decrease in reff) during the first two weeks, both in observations and in simulations. A statistically significant shift in LWP distribution is observed but not modelled in the first two weeks. A synoptic driven shift in airmass origin is observed in week-3 and attributed to the lack of observation and simulation of aerosol effects during week-3.

The manuscript is well written and easy to follow in general, but I did find some places where inherit assumptions and justifications need to be clarified. This work is intriguing in many ways and the topic is of great interest to others in the community, but I think the authors should consider/address these major points (listed below) first to make the conclusions publishable.

**Thanks for your comments on the manuscript. We have addressed the comments below to improve the manuscript.**

**Major comments:**

- A major question I have after reading this work is that where is the cloud relative to the volcanic plume, is the vertical distribution of SO2 (and its relative location to the cloud layer) changing with time, and how does this affect the cloud property changes you observed/modelled and the interpretation of them? To be more specific...

- What is eruption (injection) altitude? And what is the typical cloud top height in this region? (1)
- Are plumes always in contact with the clouds? Or there are times when they are separated? (2)
- How good is your assumption of equally distributed SO2 between 0.8 to 3 km in your ESM setup? (3)
- 1 & 3. Jordan et al. 2024 (Fig. 3a) evaluated the Holuhraun SO2 plume height retrieved from IASI satellite observations and within the UKESM1-Hol simulations used within this paper. The IASI observations show the central height of the SO2 plume exists mostly between 0.8 and 2.5 km, which is in good agreement with the UKESM1-Hol simulation volcanic SO2 emission profile that is used within this paper.

We have added the sentence "The prescribed volcanic emissions vertical profile is in agreement with satellite observations from the Infrared Atmospheric Sounding Interferometer (IASI) that show the SO2 plume height during September and October 2014 is mostly between 0.8 and 2.5 km (Jordan et al. 2024)." at L200 within the methods section to explain this agreement.

1 & 2. This is an interesting question but difficult to decipher through satellite observations. We use the OMPS satellite to retrieve the SO2 column load and create a plume mask. However, the plume height of SO2 is not available from OMPS. We have therefore investigated this comment using the UKESM1-Hol simulations and IASI observations. We have added animation S8 and S9 that compare the simulated SO2 mole fraction, IASI observed SO2 plume height and MODIS Aqua liquid cloud top height. We have also added the following text after L490 in Section 3.4.3 to explain these figures, along with additional methodology details in Section 2.2 and 2.3.

"The OMPS column amount of SO2 does not provide information on when the sulphate plume is within the cloud layer where the aerosol-cloud perturbation takes place. Therefore, we compare the SO2 plume height obtained from IASI (Carboni et al., 2016) and the height of the maximum SO2 mole fraction form the UKESM1-Hol simulations to the liquid cloud height obtained from MODIS Aqua. Animation S8 shows for each day when the SO2 plume height in the IASI observations and the height of maximum SO2 mole fraction in the Holuhraun simulations is below or above the MODIS Aqua liquid cloud top. On most days, there are grid cells within the SO2 plume that are both above and below the observed liquid cloud top height Animation S9 shows the vertical mean profile of UKESM1-Hol SO2 mole fraction, IASI SO2 plume height and MODIS liquid cloud top height when averaged over latitude. The MODIS Aqua liquid cloud top height in the latitudinal mean is close to the altitude of maximum SO2, with SO2 generally spanning above and below this height. Therefore, we expect the sulphate aerosol produced in the SO2 plume to be interacting with liquid water clouds. However, isolating when the sulphate aerosol plume is interaction with clouds is difficult to decipher and a limitation of using satellite observations alone."

- Regarding the results from UKESM simulations:
  - When interpreting/comparing modelled results to observed results, how do you address the fact that LWP adjustment due to aerosol perturbations is unidirectional by design in ESMs (i.e., only precip-suppression is parameterized, and entrainment-feedbacks are not represented), meaning modelled increase in LWP is likely exaggerated. (1)
  - What, physically, can we learn from comparing LWP in- and out-of- plume in the UKESM-Hol simulation when we know it's not representing the full chain of LWP responses? (2)
  - Have the authors considered running ensemble simulations to relax the assumption on uniformly distributed SO2 profile by varying it to create ensemble members? (3)
  - When I compare cloud properties in-plume between Hol and Ctrl simulations, I see much stronger changes (which we know is causal aerosol effect) compared to the results you get from the in- vs out- plume method; how do you reconcile this

difference? Does this mean the in vs out method is still heavily confounded by meteorological covariations? (4)

- What's the interpretation of the large difference in out-of-plume cloud properties between -Hol and -Ctrl? Aerosol effects or meteorological difference between simulations? (5)
- 1 & 2. We acknowledge that most current and previous generations of climate models represent an increase in LWP through the parameterisation of decreased auto conversion of cloud water to rain water. In the methods section we have now explicitly stated this for UKESM1: "Changes in cloud droplet number concentration (*N*d) can impact cloud droplet effective radius (Jones et al., 2001) and the autoconversion of cloud liquid water to rain water through the Khairoutdinov and Kogan (2000) scheme." (~L188).

In our analysis we assessed daily and weekly perturbations to LWP. Therefore, if satellite observations showed days when there was a decrease in LWP that was not represented in the model that could be illustrative of a model structural error – i.e. missing representation of entrainment feedbacks that reduce LWP. In our analysis of MODIS observations, most days in the first two weeks show an increase in LWP, whereas days in the 4th week are more varied (Figure 2). Our weekly analysis of UKESM1-Hol simulations shows meteorological conditions that are favourable of increases in LWP (~L466), yet we find non-significant modelled changes to LWP during these weeks. Hence, we do not appear to have a weekly case with a decrease in LWP. We have now added the more discussion on LWP and the following sentence in response the representation of decreases in LWP (~L460-480):

"the in-plume LWP in the Holuhraun simulation were not significantly greater than the values out-of-plume during weeks 1, 2 and 4 which contrasts with climate models' tendency to produce unrealistic large increase in LWP when *N*d increases (Malavelle et al., 2017, Toll et al., 2019). A weak LWP response to aerosol perturbation in UKESM1-Hol is consistent with results from HadGEM3-UKCA that is an earlier version of the aerosol-climate model used in this work (Ghan et al., 2016, Zhang et al., 2016, Malavelle et al., 2017). Ghan et al. (2016) hypothesized that the weak LWP response in HadGEM3-UKCA could be partly due to the autoconversion scheme used. If instead the meteorological conditions were favourable to the entrainment processes that can decrease LWP, we would not expect a decrease in LWP to be simulated since most current and previous generations of climate models do not include a parameterization where aerosol can impact cloud top entrainment (Toll et al., 2019)."

- 3. Thanks for the suggestion. It would be interesting to look at a model ensemble with varying vertical profiles of SO2. However, due to the computational resource require for ensemble analysis of SO2 vertical profile-induced uncertainty it is out of the scope of this study, but would be a good direction for further studies. Furthermore, Steensen et al. (2016) investigated different injection heights with a chemical transport model.
- 4 & 5. In our analysis we used a version of the OMPS  $SO_2$  mask that is the same resolution as the model to derive the plume analysis of model perturbation to cloud properties.

Figure 2 (now Figure 1) illustrates that the spatial extent of the OMPS  $SO_2$  plume is similar to the model, but it is not likely to be a perfect representation. The model is also a coarser resolution (1.875 x 1.25 degrees rather than 1 x 1 degrees) so will not explicitly resolve the observed plume structure. As such, the out of plume background in the model simulations may be more polluted than in the observations and could be a reason why the control simulation out of plume appears 'cleaner'.

We have added an extra column (middle) to Figure 2 (now Figure 1) that shows how the OMPS plume mask compares to the spatial distribution of the modelled SO2 column. We have also added additional text in the methods to explain this in more detail: "In our plume analysis of the model simulations, we use the OMPS SO2 plume mask that was created from OMPS data regridded to the model resolution." (L217). … "there may be days when there are slight differences in the spatial location of the plume and bounding box derived from observations compared to the model simulations (e.g. 25th September)." (L263).

We have also added a new figure into the SI that compares the spatial extent of the column amount of  $SO_2$  to sulphate mass concentration UKESM1-A. This figure shows that sulphate aerosol may be more widespread than the plume identified using  $SO_2$ . In response, we have added a new section (Section 3.4.3) discussing the limitations of using the column amount of  $SO_2$  to create a plume mask.

- I wonder as the eruption goes on, do you see a dilution effect of the SO2 plume, such that the background (out-of-plume) is getting more polluted with time? Does this contribute to you week-3 lack of signal in Nd? I wonder if you could show histograms of SO2 comparing in- and out-of- plume, similarly to Fig. 3?

This is an interesting point, thanks for raising. We have added a timeseries of the out-of-plume SO2 and  $N_d$  in Figure S7. This plot shows the area weighted geometric mean of out-of-plume SO2 stays below 0.2 DU with perhaps a slight increase across September 2014. In comparison, the timeseries of out-of-plume  $N_d$  shows the findings that are echoed in the paper – that the 3rd week of the eruption has higher background  $N_d$  than the other weeks in our analysis.

- Your bounding box size varies from day-to-day, meaning the degree of meteorological confounding effect also varies from day-to-day in your analysis, how do you address this issue when you compare cloud changes among days and group them into weeks? Also, since some boxes cover land, do you screen out land clouds? I think you only mention this in the caption of one of the figures, I would bring it up clearly in the methodology.

Yes, our analysis of cloud properties excludes land. In the Methods Section 2.3 we specify our analysis was focused on marine liquid clouds "We analyse marine liquid  $N_d$ ,  $r_{eff}$ , in-cloud LWP and cloud fraction." and in some of the figure captions and the discussions. This was perhaps not evident enough through the rest of the manuscript, so we have added this information in additional places - easiest to see where in the track changes document.

We vary the size of the bounding box in attempt to isolate the meteorological conditions that are most similar to those being experienced by the  $SO_2$  plume rather than using a larger area as the out of plume that would be more susceptibility to differences in meteorology (~L130). Since the spatial extent of the plume varies each day this inherently leads to differences in the sample size of the weekly data. We have now added Table S1 that shows the sample size in each week.

- I had hard time wrapping my head around the LWP responses, particularly when the statistical testing method disagrees with the mean changes between in- and out-. I understand the source of discrepancy, but I am a bit concerned about a lack of high-level take-away of these results in the context of ACI, i.e., should we take away with the message that LWP response is weak and its sign is undiscernible? Personally, I prefer the statistical testing method that focuses on the distribution shift rather than a change in the mean, which could be driven by outliners and not representing a physical response.

We agree that the differing LWP result between the statistical test and the mean inplume enhancement in the original paper is confusing for a high-level takeaway. Following this comment, we have decided to use the area-weighted geometric mean instead in our calculation of in-plume enhancement throughout the paper and in the values within Table 1. The geometric mean is more appropriate for skewed distribution. We previously calculated the area weighted mean of in and out-of-plume values each day and then the arithmetic means to get the weekly mean. The in-plume enhancement for LWP calculated using the geometric mean is in the same direction as the statistical tests results, and we hope this improves the high-level takeaways of the paper for the reader. In addition, throughout the paper we mainly refer to the results of the statistical tests when discussing the results.

- Regarding the satellite retrieved cloud properties,
  - Weren't you concerned about getting mixed-phase or ice clouds when your 1-5km cloud-top constraint is well above the freezing level? (1)
  - For Nd retrievals, high-SZA and low-CF have been shown to produce unreliable retrievals (e.g., Grosvenor et al. 2018), I would put extra constraints on these two variables. (2)
  - Does SO2 plume in the scene affect satellite microphysical (tau and reff) retrievals?
     (3)
  - Our analysis focuses only on liquid clouds as retrieved from the MODIS phase retrieval algorithm. We describe this in the methods (L148-151): "We analyse marine liquid Nd, reff, in-cloud LWP and cloud fraction ... Cloud phase is retrieved through the phase retrieval algorithm at 1 km resolution." We acknowledge that there are uncertainties regarding the phase retrieval algorithm which is why we previously applied the additional cloud top height constraint. Cloud top height was not available from the L3 COSP dataset and we have therefore removed this constraint.
  - 2. We have not applied the extra constraint on cloud fraction when calculating  $N_d$  for consistency with other studies of the Holuhraun eruption (e.g. Chen et al. 2022). While cloud fraction is not constrained, our focus is not the absolute value of  $N_d$  but the change of  $N_d$  from non-polluted to polluted conditions, which are expected to

experience similar bias and therefore only marginally influence our analysis of aerosol's impacts on clouds. Our OMPS analysis excludes pixels where SZA > 70 degrees and consequently higher latitudes from the MODIS dataset that are less reliable are excluded as September progresses.

3. In the MODIS COSP dataset, cloud properties are estimated for pixels identified as confidently or probably cloudy. Pixels identified as cloudy are excluded from MODIS COSP retrievals if multi-spectral tests suggests that they are sunglint or heavy aerosol (Pincus et al. 2023).

**Minor comments:**

• What's the main reason and benefit for extending analysis to 4 weeks? Are you targeted to capture temporal evolution in ACI and/or its timescale? Or just want to explore the influence of different synoptic/meteorological patterns? If it's the latter, why only one month is analyzed when you can do this for the whole eruption time period? Why group your analysis by week (is there any physical reason? Timescale assumption? Any major synoptic pattern shift at weekly scale)? I think these inherit assumptions and justifications need to be layout upfront clearly.

We have added additional information to the last paragraph of the introduction to justify why we use 4 weeks in September 2014 for our analysis (~L96-100):

"The eruption was at its most powerful in September 2014 with large amounts of SO2 released that then reduced during October 2014 (Carboni et al., 2019). The 4week time period allows us to investigate how airmass history and background meteorological factors influence aerosol-cloud interactions between the weeks of our analysis using the HYSPLIT trajectory model (The Hybrid Single-Particle Lagrangian Integrated Trajectory model). A week-by-week analysis is performed showing that the aerosol conditions in the first two weeks and the last week of September are close to pristine, but during the third week, the background aerosol is significantly perturbed owing to airmass trajectories originating over continental Europe. This breakdown into weeks provides a convenient framework for developing statistical analyses over the month."

It is serendipitous that the ACI behaviour across our analysis fits well into the weeks of September (e.g. now Figure 2) and convenient to look at the meteorology when grouped by weeks. We mention this at ~L303:

"To investigate the lack of perturbation to the in-plume  $N_d$  for many days of the 3rd week of September and why there is a variation in the in-plume LWP response across September, we aggregate our daily plume analysis into the weeks of September."

• Information on UKESM initialization and boundary conditions needs to be added. Following this comment and that of Reviewer 2 we have added additional information on the description UKESM simulations in Section 2.4 – please see track changes.

- Need to mention why there are missing days in the analysis (e.g., Fig. 4). We have added the following sentence to L122: "The OMPS SO2 vertical column density is unavailable 1 day each week and we exclude these dates from our analysis."
- Line 87, check sentence "... to and ..." Thanks for spotting this typo. We have changed it to "...and an..."
- Figures, missing lat/lon labels when maps are shown, and please add color code to the caption.
   We have added lat/lon coordinates labels the relevant figures in the main paper.
- Line 213, "e.g." should be inside the parenthesis. Changed so e.g. is inside the figures.
- Line 226, I think Sep-11 and Sep-25 in Fig. 2 are examples where they do not agree, please discuss and reword. Line 231, you meant "north" of the domain? as "top" refers to the vertical direction.
   In line with a main comment above we have altered the text in the paragraph to "...but there may be days where there are slight differences in the spatial location of the plume and bounding box derived from observations compared to the model

simulations (e.g. 25th September)"

- Line 270-271, after reading through the whole manuscript, I am still missing an explanation on week-4 LWP responses, observed and modelled. Using the updated methodology our observed and modelled LWP responses in Week 4 are in line with Weeks 1 and 2. We have added more discussion on LWP responses at L560-575.
- I feel Fig. S2 is worthy of being included as a main figure. Thanks for the suggestion. Figure S2 is now included as Figure 4 in the main text.
- Line 303, what is the "reason" that you're referring to? And how do you know it's necessarily the same reason in the simulation, rather than different reasons leading to the same results? perhaps need to reconstruct this sentence.
   We have removed this part of the sentence as this paragraph discusses results and not potentially mechanisms for similarity/differences between MODIS and UKESM.
- Table 1, cloud fraction responses in simulations? Line 328, perhaps better to define perturbation in the main text and use a mathematical expression.
   We have added the cloud fraction responses to the table. See list of main changes above.
- Line 344-347, this discussion in not very clear and hard to follow, especially the use
  of terms like 'updraft-limited' and 'aerosol-limited', which are not straightforward
  concepts and need to be explained/introduced.
  Thanks for this feedback. We have added a more in-depth description about the
  activation of aerosols into cloud droplets and introduce the regimes of aerosol
  activation earlier in the paper when it is first mentioned:

"The activation of the Holuhraun aerosol plume into cloud droplet depends on multiple factors. These factors include the number, size and hygroscopicity of aerosol particles, as well as the updraft velocity at cloud base and the water vapour supersaturation. Reutter et al. (2009) showed the activation of aerosol into cloud droplets can occur under three regimes: updraft-limited, aerosol-limited or aerosol- and updraft-sensitive regimes. The updraft-limited activation regime is characterized by low ratios of updraft velocity/aerosol number concertation and hence is more likely to occur under polluted air masses, such as week 3 in our analysis (Jones et al., 1994; Reutter et al., 2009; Carslaw et al., 2013; Spracklen and Rap, 2013). In this updraft-limited regime the activation of aerosol to cloud droplets depends on updraft velocity rather than aerosol concentration. As a result, under this regime, polluted air masses arriving in the region of the Holuhraun aerosol plume during the 3rd week would be less susceptible to further aerosol-induced increases in Nd. In comparison, in the aerosol-limited region the activation of aerosol to cloud droplets is proportional to the aerosol number concentration."

• Line 365-371, why choose modelled meteorological conditions instead of ERA reanalysis? And define how is LTS calculated.

At L412 we explain why we use the modelled meteorological conditions: "Since there is agreement between the lack of ACI signal in observations and simulations in the 3rd week, we use the UKESM1-Hol simulation to investigate differences in meteorological conditions during the 3rd week that may contribute towards the negligible in-plume aerosol perturbation to cloud properties."

In addition, there may be biases between ERA meterological condition and those simulated by UKESM1. But, we have now added here that the model simulation are nudged to ERA reanalysis at L421:

"The model simulations are nudged to ERA-Interim reanalysis horizontal winds and potential temperatures."

And how we calculate LTS at L427:

"We calculate LTS as the difference in potential temperature between 720 and 1000 hPa."

- Lines 412-414, any speculation on why this is the case? Ghan et al. (2016) hypothesized that the weak LWP response in HadGEM3-UKCA is partly due to the autoconversion scheme used. We have added this into the discussion on meteorological variables at ~L472: "A weak LWP response to aerosol perturbation is consistent with results from HadGEM3-UKCA that is an earlier version of the aerosol-climate model used in this work (Ghan et al., 2016, Zhang et al., 2016, Malavelle et al., 2017). Ghan et al. (2016) hypothesized that the weak LWP response in HadGEM3-UKCA could be partly due to the autoconversion scheme used."
- Lines 484-486, I think this needs to be mentioned upfront, I have been wondering about CF responses when I read the results.
   Since, we have now added the model-observation cloud fraction we have added an additional paragraph to the discussion on the cloud fraction response ~L576-584:
   "Chen et al. (2022) showed a significant increase in satellite observations cloud fraction following the Holuhraun eruption when using a machine learning approach

that accounts for meteorological confounders. Consistently, our results show an observed increase in cloud fraction during the first two weeks of September 2014. In the first week the increase is simulated by the volcanic and control UKESM1 simulations, although the increase in cloud fraction is larger in the volcanic simulation. However, in the second week the simulations show a decrease in cloud fraction. In the fourth week, there is a non-significant decrease in observed cloud fraction but a significant decrease in the model simulations. The similarity in the inplume perturbation to cloud fraction between the volcanic and control simulations across our analysis indicates much of the simulated cloud fraction change is likely dominated by meteorological covariability. Further simulations would be needed to isolate if the smaller differences between the in-plume perturbation to cloud fraction simulations. ""

**References**

Grosvenor, D. P., Sourdeval, O., Zuidema, P., Ackerman, A., Alexandrov, M. D., Bennartz, R., Boers, R., Cairns, B., Chiu, J. C., Christensen, M., Deneke, H., Diamond, M., Feingold, G., Fridlind, A., HÃŒnerbein, A., Knist, C., Kollias, P., Marshak, A., McCoy, D., Merk, D., Painemal, D., Rausch, J., Rosen- feld, D., Russchenberg, H., Seifert, P., Sinclair, K., Stier, P., van Diedenhoven, B., Wendisch, M., Werner, F., Wood, R., Zhang, Z., and Quaas, J.: Remote Sensing of Droplet Number Con- centration in Warm Clouds: A Review of the Current State of Knowledge and Perspectives, Rev. Geophys., 56, 409–453, https://doi.org/10.1029/2017RG000593, 2018.

Steensen, B. M., Schulz, M., Theys, N., and Fagerli, H.: A model study of the pollution effects of the first 3 months of the Holuhraun volcanic fissure: comparison with observations and air pollution effects, Atmos. Chem. Phys., 16, 9745–9760, https://doi.org/10.5194/acp-16-9745-2016, 2016.

Pincus, R., Hubanks, P. A., Platnick, S., Meyer, K., Holz, R. E., Botambekov, D., and Wall, C. J.: Updated observations of clouds by MODIS for global model assessment, Earth Syst. Sci. Data, 15, 2483–2497, https://doi.org/10.5194/essd-15-2483-2023, 2023.

**Review 2**

**Review of "In-plume and out-of-plume analysis of aerosol-cloud interactions derived from the 2014-15 Holuhraun volcanic eruption" by Peace et al., submitted to Atmospheric Chemistry and Physics (ACP) Manuscript number: "acp-2024-360" Decision: "Major revision"**

The study focuses on the uncertainty surrounding aerosol effective radiative forcing (ERF) and aerosol-cloud interactions, which are crucial for understanding climate sensitivity and predicting future climate change. Using the 2014-15 Holuhraun volcanic eruption as a natural experiment, the researchers evaluate the impact of volcanic aerosols on cloud properties during the first month of the eruption, comparing observations with simulations from the UK Earth System Model (UKESM1-A). During the initial two weeks of the eruption, both observations and simulations show a shift to smaller and more numerous cloud droplets within the volcanic plume, along with changes in liquid water path (LWP) values. However, in the third week, this shift is neither observed nor accurately modeled, and discrepancies exist between observations and simulations in the fourth week. The study underscores the influence of air mass history and background meteorological factors on aerosol-cloud interactions across different weeks. Most parts of the manuscript are well written, but there are several issues to be addressed. Based on the descriptions outlined above, my decision is "major revision," and I encourage the authors to revise the manuscript.

Thank for your comments and review of the manuscript. We have added some additional analysis and discussion in response to the comment which improves the manuscript.

**Major comments:**

**Definition of Volcanic Plume:**

Using SO2 as a proxy for the volcanic plume in the 3rd and 4th weeks following an eruption may not be entirely accurate, as the SO4 formed from it can dissipate from the source and the location of the SO2 plume and SO4 plume could be different. Therefore, it is possible that in the bounding boxes outside the plume area, some fingerprints of SO4 could still be present. Have you thought about comparing the SO4 plume with the SO2 plume in your volcano simulations? I recommend providing a figure similar to Figure 2 for UKESM1-A, but with plots for SO4. Or alternatively provide a comparison in distribution of one of the variables such as cloud droplet effective radius for outside of plume in simulations with the volcano eruption.

This is a good point, thank you for raising. Following this comment we have repeated the analysis we did for UKESM1-Hol  $SO_2$  with the vertical mean  $SO_4$  mass concentration. We have added the  $SO_4$  figure and animation into the supplementary.

We have added Section 3.4.3 that describes the limitations of using the observed column amount of  $SO_2$  to identify with  $SO_4$  interacts with clouds:

"In our analyses we use the column amount of SO2 to track the aerosol plume as this information is readily available from satellite observations and model simulations. We assume that the column amount of SO2 is a good proxy for where sulphate aerosol is produced, as this information is not observable from satellite observations. Figure S1 shows how the column amount of SO2 compares to the vertical mean sulphate mass concentration in the UKESM1-Hol simulations. The spatial location of sulphate aerosol is in good agreement with the location of the column amount of SO2 for our snapshot days. However, the unmasked sulphate mass concentration is elevated across a larger area both inside and outside of the plume mask bonding box. The more widespread enhanced aerosol load revealed by the sulphate mass concentration, in combination with slight differences between the modelled and observed SO2, is likely why the out-of-plume  $N_d$  in the UKESM1-Hol concentration is larger than in the UKESM1-Ctrl. The absolute values of  $N_d$ observed by MODIS are lower than in UKESM1-Hol, and the MODIS out-of-plume  $N_d$  is comparable to the out-of-plume  $N_d$  in UKESM1-Ctrl."

**UKESM1-A simulations:**

I believe that additional details regarding your simulations are necessary. Could you please provide information about the cloud microphysics, cloud cover, and convection scheme utilized in your simulations?

- Additionally, since you use the satellite simulator from the COSP package, were subcolumns employed, and if so, how many subcolumns were utilized given the coarse resolution which is used?
- I'd like to inquire whether the information of size distribution of hydrometeors used as an input for the MODIS simulator, which is essential for simulating MODIS signals, was taken into account in your simulations?
- Regarding the resolution of your simulations, which is mentioned as 1.875 x 1.25° at the equator.
- What does this resolution correspond to in the North Atlantic, where your analysis takes place? Additionally, it's noted that OMPS and MODIS data are gridded at 0.5 x 0.5-degree resolution. Could you please explain how this resolution compares to your simulation's resolution?

**We have added additional details on the set up of UKESM1 in response to points 1 and bullet points 3 and 4 in this comment. The model description (L180-200) now includes:**

"UKCA uses aspects of the Unified Model Global Atmosphere (GA7.1; Walters et al., 2019) within the UKESM for the large-scale advection, convective transport and boundary layer mixing of aerosol. Aerosol particles are activated into cloud droplets using Abdul-Razzak and Ghan (2000) activation scheme. Large-scale cloud microphysics is a single-moment scheme based on Wilson and Ballard (1999) with improvements based on Boutle et al. (2014). Changes in cloud droplet number concentration (*N*d) can impact cloud droplet effective radius (Jones et al., 2001) and the autoconversion of cloud liquid water to rain water through the Khairoutdinov and Kogan (2000) scheme. Aerosol–cloud interactions are simulated in large-scale liquid clouds. Convection is parameterized separately to large-scale clouds and does not consider aerosol. Bulk properties of large-scale clouds are simulated using the prognostic cloud fraction and prognostic condensate (PC2) scheme (Wilson et al., 2008a, b) with the modification described in Morcette (2012). The GA7.1 model and its coupling to UKCA utilised are described in further detail in Walters et al.

**(2019) and Mulcahy et al. (2020)." ...**

"We use global model simulations with a resolution of N96L85, which is a horizontal resolution of  $1.875 \times 1.25^{\circ}$  (~208 × 139 km at the equator and ~86 x 139 km near the Holuhraun eruption), with 85 atmospheric levels. The model resolution is coarser than the MODIS and OMPS datasets we use that are at  $1.0 \times 1.0^{\circ}$  resolution."

Regarding bullet points 1 and 2. For the purpose of the review, the COSP implementation in the Unified Model (UM) uses 64 subcolumns which is the number of subcolumns used by the cloud generator in the radiative transfer code. The inputs to the MODIS simulator are described in Pincus et al. (2012).: "The MODIS simulator requires a greater diversity of inputs than does the ISCCP simulator, including profiles of particle size for liquid and ice clouds ... and the corresponding liquid and ice optical depths at 0.67 µm within each layer of each subcolumn as a function of the model's vertical coordinate z. Users may opt to provide a single value of optical depth and the mixing ratios of cloud ice and liquid, in which case optical depth is partitioned by phase, assuming that particles are in the geometric optics limit.". We think this is too in depth to add in our methods section but have now cited the Pincus et al. 2012 reference.

The LWP response:

- In your paper, there is significant discussion regarding the LWP response. I believe it is essential to include analyses of LWP for different weeks of the study in the main manuscript. Therefore, I suggest providing a figure similar to Figure 5 for LWP.
- Does your analysis of precipitation in Figure 8 for the first two weeks of the eruption indicate any suppression in precipitation?

**Thanks for the suggestion. We have added the updated LWP figure to the main paper (now Figure 4).**

We had a look if there was any precipitation suppression evident in the simulations creating the bar plot for in and out-of-plume precipitation, but none was evident. In doing so, we realised we had not converted the precipitation units to mm day-1 as per the y axis so have updated the units in the meteorological variable plot.

Discussion about cloud fraction:

- Could you elaborate on why analyses in Table 1 for cloud fraction are not included for simulations?
- It would improve the discussion on cloud fraction to compare the results obtained from MODIS with those presented in the study by Chen et al. (2022), which suggests enhancing cloud fraction appears to be the leading cause of climate forcing.

This is a good point about cloud fraction. Following the reviewer comments we have added cloud fraction for the model simulations to Table 1. As discussed in the major changes part of the response we have updated the MODIS dataset to the L3 MODIS COSP dataset and changed the cloud fraction variable to liquid cloud retrieval fraction, which is more consistent (in terms of retrieval algorithm) with the liquid cloud retrieval microphysical properties.

We have also added discussion around our liquid cloud fraction results. For example at L575-590:

"Chen et al. (2022) showed a significant increase in satellite observations cloud fraction following the Holuhraun eruption when using a machine learning approach that accounts for meteorological confounders. Consistently, our results show an observed increase in cloud fraction during the first two weeks of September 2014. In the first week the increase is simulated by the volcanic and control UKESM1 simulations, although the increase in cloud fraction is larger in the volcanic simulation. However, in the second week the simulations show a decrease in cloud fraction. In the fourth week, there is a non-significant decrease in observed cloud fraction but a significant decrease in the model simulations. The similarity in the in-plume perturbation to cloud fraction between the volcanic and control simulations across our analysis indicates much of the simulated cloud fraction change is likely dominated by meteorological covariability. Further simulations would be needed to isolate if the smaller differences between the in-plume perturbation to cloud fraction in the control and Holuhraun simulations could be attributed to aerosols. For example, Grosvenor and Carslaw (2020) examined the contributions of changes in  $N_d$ , LWP and cloud fractions to pre-industrial to present-day aerosol ERF in UKESM1-A. Their results showed that LWP and cloud fraction were the dominant terms in the radiative forcing of aerosol-cloud interactions over the North Atlantic, and that cloud fraction changes are more dominant in regions of broken cloud. An additional simulation was conducted in the Grosvenor and Carslaw (2020) study where Nd was prevented from modifying rain formation through the autoconversion parameterisation, in these siumulations there was a negligible change in cloud fraction over the North Atlantic."

**Minor comments:**

- In Table 1, can you please explain why the mean value for Nd for outside plume in control and Hol simulations for week 3 and 4 is very different? I believe this also can be related to my first comment on the definition of volcano plume.
   We have added discussion on this see response to the main comment above regarding definition of a volcanic plume.
- Do the simulations conducted with UKESM1-A cover the entire globe? It would be beneficial to briefly discuss the advantages and disadvantages of employing global simulations compared to regional cloud-resolving simulations. Yes, the Holuhraun simulations are global model simulations as noted in the methods section ~L195: "We use global model simulations with a resolution of N96L85..." We note that additional model simulations such as those with a nested regional model would be useful at the end of the paper.
- Please provide some short information in the manuscript regarding the time at which you analyzed simulations. Is it the daily mean or at the time of MODIS-AQUA overpass?
   As described in the major changes section of the response, we have updated our analysis to use the MODIS COSP L3 daily dataset that combines pixel-scale observations from Terra and Aqua to a 1x1 degree grid. From this dataset, "we use the mean of the sampled Level 2 pixels in each Level 3 grid" (~L147).

• In Figure 5, regarding the average enhancement observed for reff and Nd, I'm curious about why the decrease in mean enhancement for reff is more pronounced in weeks 4 and 2 compared to week 1, while the increase in mean enhancement in Nd is less pronounced in weeks 2 and 4 compared to week 1. Regarding the fact that in the relationship used to calculate Nd, it seems that reff has a stronger impact compared to cloud optical depth.

Using the updated datasets and methodology for calculating the in-plume mean enhancement (as described in major changes section of response) we now find that both the increase in Nd and decrease in reff is largest in Week 1, followed by week 2 and week 4.

- I propose moving Figure 3 to the supplementary materials or combining the information in the legend with Figure 5. We have moved this figure to the supplementary materials.
- In line 191, did you mean event by vent? In this instance, the vent refers to the grid cell in which the model emissions are prescribed.
- In Figure 4 for reff, it is demonstrated in< out the plume while in figure 5, for reff, the in> out plume is demonstrated. I recommend maintaining consistency between these two figures.
   Thanks for spotting this. It was a typo in our code and has now been corrected to be consistent.
- In the abstract, can you discuss briefly how LWP has changed in the first 2 weeks? In the current version, it is just mentioned that it is changed.
   We have updated the abstract to include the increase in observed LWP in these weeks that is not replicated by the model. "We find an observed increase liquid water path (LWP) values inside the plume that is not captured in UKESM1"
- In Figure 5 caption, Nd should be Nd. Corrected, thanks.

**References**

Pincus, R., S. Platnick, S. A. Ackerman, R. S. Hemler, and R. J. Patrick Hofmann, 2012: Reconciling Simulated and Observed Views of Clouds: MODIS, ISCCP, and the Limits of Instrument Simulators. J. Climate, 25, 4699–4720, https://doi.org/10.1175/JCLI-D-11-00267.1.

---

## Referee Report (RR1)

Review of the manuscript numbered acp-2024-360, revised version 1
Title: "In-plume and out-of-plume analysis of aerosol-cloud interactions derived from the 2014-15 Holuhraun volcanic eruption"
written by Amy H. Peace et al.
Manuscript number: "acp-2024-360".
Decision: "Minor revision"

The authors have done a good job responding to my concerns and comments. I am thankful for the authors' efforts to revise the manuscript. However, I have some minor comments therefore I suggest a minor revision. I use the section numbers and page numbers based on the revised manuscript.

- In Section 2.3, could you please provide explanations on how MODIS COSP Level 3 differs from standard traditional MODIS L3 data, and why it is necessary to use this specific dataset?
- In the caption of Figure 1, could you please clarify the difference between OMPS and MOPS-coarse?

- In the caption for Table 1, "Nd" and "d" should be corrected.
- Line 430: You mention Figure 8, but the paper does not contain a Figure 8.
- These citations are missing DOIs:
    Jones, A., Roberts, D. L., Woodage, M. J., and Johnson, C. E.: Indirect sulphate forcing in a climate model with an interactive sulphur cycle, J. Geophys. Res., 106, 20293–20310, 2001
    Khairoutdinov, M. F. and Kogan, Y. L.: A new cloud physics parameterization in a large-eddy simulation model of marine stratocumulus, Mon. Weather Rev., 128, 229–243, 2000.
- The year should be at the end here:
    Pincus, R., S. Platnick, S. A. Ackerman, R. S. Hemler, and R. J. Patrick Hofmann, 2012: Reconciling Simulated and Observed Views of Clouds: MODIS, ISCCP, and the Limits of Instrument Simulators. J. Climate, 25, 4699–4720, https://doi.org/10.1175/JCLI-D-11-00267.1.

- A dot is missing here at the very end:
    Trofimov, H., Bellouin, N., and Toll, V.: Large-scale industrial cloud perturbations confirm bidirectional cloud water responses to anthropogenic aerosols. Journal of Geophysical Research: Atmospheres, 125, e2020JD032575. https://doi.org/10.1029/2020JD032575, 2020

---

## Author Response (AR2)

**Response**

We would like to thank the reviewers for their time reviewing the changes we made to the manuscript.

We have addressed the below comments in Report #2 as follows:

- In Section 2.3, could you please provide explanations on how MODIS COSP Level 3 differs from standard traditional MODIS L3 data, and why it is necessary to use this specific dataset?
  We have added the following text ~L150 to describe how MODIS COSP Level 3 differs from standard Level 3:
  "... The dataset was recently produced to facilitate comparison with results from the COSP (CFMIP Observation Simulator Package) MODIS simulator that is a software tool that can be employed in climate models to produce data comparable to satellite observations. The definitions of variables within this dataset are more in line with the MODIS simulator than standard MODIS products. Therefore, the MODIS COSP dataset is particularly useful for observation-model comparison."

- In the caption of Figure 1, could you please clarify the difference between OMPS and MOPS-coarse?
  We have clarified the differences between OMPS and OMPS-coarse in Figure 1 caption:
  "Figure 1: Total column amount of $SO_2$ (Dobson Units) retrieved from OMPS (1.0 x 1.0 °), OMPS-coarse (OMPS regridded to UKESM1-Hol resolution) and simulated in UKESM1-Hol within the plume mask for the midweek day of the four weeks in September 2014 being analysed."

- In the caption for Table 1, "Nd" and "d" should be corrected.
  Corrected.

- Line 430: You mention Figure 8, but the paper does not contain a Figure 8.
  This should have referred to Figure 7 and has now been corrected.

- These citations are missing DOIs:

  Jones, A., Roberts, D. L., Woodage, M. J., and Johnson, C. E.: Indirect sulphate forcing in a climate model with an interactive sulphur cycle, J. Geophys. Res., 106, 20293–20310, 2001

  Khairoutdinov, M. F. and Kogan, Y. L.: A new cloud physics parameterization in a large-eddy simulation model of marine stratocumulus, Mon. Weather Rev., 128, 229–243, 2000.

The year should be at the end here:

Pincus, R., S. Platnick, S. A. Ackerman, R. S. Hemler, and R. J. Patrick Hofmann, 2012: Reconciling Simulated and Observed Views of Clouds: MODIS, ISCCP, and the Limits of Instrument Simulators. J. Climate, 25, 4699–4720, https://doi.org/10.1175/JCLI-D-11-00267.1.

A dot is missing here at the very end:

Trofimov, H., Bellouin, N., and Toll, V.: Large-scale industrial cloud perturbations confirm bidirectional cloud water responses to anthropogenic aerosols. Journal of Geophysical Research: Atmospheres, 125, e2020JD032575. https://doi.org/10.1029/2020JD032575, 2020

We have corrected the citation formatting.

- In addition to the above technical corrections, we spotted a couple more. We have corrected the following:

  At L210: model simulations are just nudged to winds not wind and temperature.

  Code and data availability: was missing L3 MODIS COSP source and we have added in the Zenodo doi that was a placeholder previously.